# The Lasso and the Factor Zoo-Predicting Expected Returns in the Cross-Section

Marcial Messmer and Francesco Audrino *

Department of Economics, School of Economics and Political Science, University of St. Gallen, Bodanstrasse 6, 9000 St. Gallen, Switzerland
* Correspondence: francesco.audrino@unisg.ch

**Abstract:** We investigate whether Lasso-type linear methods are able to improve the predictive accuracy of OLS in selecting relevant firm characteristics for forecasting the future cross-section of stock returns. Through extensive Monte Carlo simulations, we show that Lasso-type predictions are superior to OLS when type II errors are a concern. The results change if the aim is to minimize type I errors. Finally, we analyze the predictive performance of the competing methods on the US cross-section of stock returns between 1974 and 2020 and show that only small and micro-cap stocks are highly predictable throughout the entire sample.

**Keywords:** factor models; cross-section of stock returns; lasso; simulation study

## 1. Introduction

After years of strong growth in the number of published firm characteristics (FC) claiming to explain differences in average cross-sectional returns, some researchers have more recently shifted their attention to the fundamental question of which statistical method to employ in selecting these variables; see for example, Harvey et al. [1], McLean and Pontiff [2] or Green et al. [3]. Given that understanding differences in cross-sectional returns has far-reaching implications for finance theory in general and consequently also for a vast part of the investment management industry, improving these methods is a pre-requisite for future finance research. This work aims to contribute to the task by investigating the importance of selecting FC that matter for prediction in a selection process focusing on prediction and highlighting the relative predictive accuracy of various shrinkage methods through an extensive simulation study and an empirical investigation of cross-sectional returns in the US.

More generally, selecting variables, estimating coefficients and predicting noisy targets are common challenges for finance and economics. An important application in the context of selecting FC is the seminal contribution by Fama and French [4], where variable selection is performed based on the multivariate regression framework and where insignificant coefficients are discarded. In particular, they regress cross-sectional returns on several firm characteristics to determine the crucial set of criteria that explain differences in returns. Based on this selection procedure, Fama and French [5] form the well-known Fama-French (FF) three-factor model, which has set the benchmark and raised the bar for detecting new relevant FC. However, these estimates, usually obtained from ordinary least squares (OLS), often suffer from a large variance and, hence, conclusions about the relevance of coefficients come potentially with a high degree of uncertainty.

To overcome the high variance problem of classical linear methods, the machine learning literature has introduced alternative methods for variance reduction by tolerating a small bias. In an important contribution Tibshirani [6] presents the least absolute shrinkage and selection operator (Lasso) method for estimating linear models. It simultaneously

performs variable selection and coefficient estimation by shrinkage. To preserve the advantages of absolute shrinkage, Zou [7] proposes a modified version, the so-called adaptive Lasso, such that consistent variable selection can be achieved even under less stringent conditions.

This study contributes to the literature by developing an extensive Monte Carlo simulation to generate a panel of plausible cross-sectional returns in which a distinct and novel feature is the flexible simulation of high-dimensional FC correlation matrices. This simulation design allows us to investigate extensively the predictive performance of Lasso methods in panels for various error specifications and to highlight eventual problems related to the correct selection of FC that contain useful information to predict the cross-section of expected returns.

The primary goal of the paper is to answer the question of whether Lasso-type methods can be useful in predicting differences in expected cross-sectional returns. Secondly, the paper aims to determine which firm characteristics drive these predictions and how they compare to classical approaches. In addition to the empirical evaluation, we use a simulation study to shed light on the properties of the methods in finite samples. For the empirical part of our analysis, we focus on the US cross-section. We include 62 published firm characteristics constructed based on the CRSP/Compustat merged database with monthly data starting from 1974 until 2020.

It is important to note that we constrain our selection and prediction procedure to the linear setting. Specifically, we want to perform our prediction based on a multivariate regression that is consistent with the original scope when the considered FC was introduced in the literature. Hence, this study builds on the work of Green et al. [3]. In particular, the authors analyze a large set of FC in a linear multivariate Fama and MacBeth [8] regression; we closely follow their data construction and FC pre-selection procedure. However, instead of relying on a multivariate regression, we apply the adaptive Lasso a true variable selection method.

The simulation results indicate some advantages of the adaptive Lasso over the Lasso in selecting the true set of FC. In contrast, Lasso-type predictions rank consistently better when predictive accuracy is the main objective. We find patterns consistent with the simulation results when predicting US small-cap stock returns, as two of the considered Lasso-type specifications achieve the best predictions. Large-cap stocks are not forecastable with the methods included in this work and the naïve zero return forecast cannot be rejected as being inferior to the included set of linear estimators. These results on the US expected returns cross-section confirm and extend the empirical evidence provided in the previous literature.

The full pooled panel adaptive Lasso selection characterizes 21 FC of relevance for future differences in stock returns. This is in stark contrast to 47 variables selected by the Lasso, 23 by pooled ordinary least squares (POLS) and 13 by POLS inference corrected for multiple testing. The most dominant FC for prediction is based on price information; the most consistently selected is short-term reversal. Moreover, the Fama and French [9] five-factor model is fully represented in the Lasso-based selection, but complemented by additional FC. Although the methods considered in the current study substantially differ, generally this contrasts with the findings of Green et al. [3], as they identify a relatively low-dimensional linear cross-section.

This study contributes to different strands of the literature. First, it contributes to the asset pricing literature by analyzing the usefulness of Lasso-type methods in selecting relevant FC for estimating and predicting expected cross-sectional stock returns; we refer, among others, to Cochrane [10,11], Goyal [12], and Hou et al. [13] for reviews of the different research questions, estimation methods, and introduced FC related to asset pricing. Harvey et al. [1] introduce the concepts of family-wise error and the false discovery rate to the finance literature. Applying the t-value adjustment reveals that many published factors would lose their status as a significant factor. However, the method suffers shortcomings from a prediction perspective that our work takes into consideration: It does not explicitly

take into account the dependence structure of the FC and it neglects to trade-off type I vs. type II errors.

More recently, Kozak et al. [14] investigate the problem from a portfolio perspective in combination with $L_2$ and $L_1$ penalties. The authors identify a sparse set of FC. A mean-variance (MV) optimized portfolio including 50 anomaly variables yields a CAPM alpha similar to the Fama and French [9] five-factor model. Furthermore, the authors include two-dimensional interactions between these 50 FC and show a substantial increase in an alpha of the MV portfolio compared to the case without interactions.

Feng et al. [15] propose a double Lasso model selection methodology to systematically investigate the in-sample marginal contribution to asset pricing of some new, additional factors beyond what is explained by a possibly vast number of already existing ones. They introduce a framework for conducting in-sample statistical inference in such a high-dimensional setting and provide robustness checks to verify the sensitivity of the results with respect to the involved tuning parameters in finite samples. In contrast to their study, our analysis focuses on out-of-sample prediction and on the evaluation of the forecasting accuracy of the resulting Lasso-based models. We provide new evidence about the finite sample properties of the Lasso (and other) estimators to select relevant factors for prediction through an extensive simulation exercise that is broader in terms of competing models and model selection criteria designed for forecasting than the one presented by Feng et al. [15].

Finally, Bryzgalova [16] pays particular attention to problems arising from model mis-specification when using shrinkage methods in the context of factor models. She introduces an alternative adaptive weighting scheme based on partial correlations instead of a two-stage procedure as compared with the adaptive Lasso. The work of Freyberger et al. [17] approaches the problem using non-parametric techniques. DeMiguel et al. [18] analyze the FC selection from a portfolio perspective in a framework that combines shrinkage and mean-variance (MV) optimization. Moreover, a fast-growing strand of the literature addresses the prediction problem from a non-linear perspective; see, for example, Messmer [19], Moritz and Zimmermann [20] or Gu et al. [21].

Second, our study is related to the literature that investigates the finite samples or asymptotical properties of shrinkage approaches in financial settings. The Lasso introduced by Tibshirani [6] is motivated by the desire to improve OLS estimates without the short-comings of subset selection and ridge regression. Tibshirani [6] notes that subset selection suffers from high variability, as small data changes can cause subset selection to easily select a different model. Zou [7] remarks that subset selection can become computationally infeasible if the number of variables is large. Ridge regression, which penalizes the sum of the squared coefficients ($l_2$ norm) in a linear regression framework, on the other hand, has no obvious interpretation due to the fact that the coefficients are not exactly set to zero.The Lasso estimator optimizes least squares under an additional condition involving the total sum of the absolute size of the coefficients (known as the $\ell_1$-norm) that cannot be larger than a given tolerance value. The inclusion of a penalty term leads to consistent coefficient estimation and variable selection if two necessary conditions are fulfilled, as Meinshausen [22] shows; see Bühlmann and Van De Geer [23] for a detailed discussion. These conditions are too restrictive in many empirical applications. Zou [7] modifies the Lasso insofar as the weight of each coefficient in the penalization term is adaptive. This is achieved by scaling the absolute value of each coefficient with a first-stage estimator such that more highly relevant variables are less strongly affected by the penalty. Setting adaptive weights leads to consistent variable selection and coefficient estimation even if one of the two is not fulfilled.

The previously mentioned consistency properties are developed for a cross-sectional set-up with iid errors. Typically, the majority of applications in finance require the use of time-series or panel data. Moreover, an iid error specification is more an exception than the rule. Consequently, Medeiros and Mendes [24], Caner and Zhang [25], Caner and Kock [26], Kock and Callot [27], Audrino and Camponovo [28] and Kock [29,30] derive asymptotic properties of the Lasso and the adaptive Lasso in time series and panel

settings. In particular, Medeiros and Mendes [24] and Audrino and Camponovo [28] derive consistency properties of the adaptive Lasso in time series environments. Medeiros and Mendes [31] prove that the oracle properties of the adaptive Lasso are preserved for linear time series models even under non-Gaussian, conditionally heteroscedastic and time-dependent errors. Audrino and Camponovo [28] show that the adaptive Lasso combines efficient parameter estimation, variable selection and valid finite sample inference for general time series regression models. We contribute to this strand of the literature by investigating the finite sample properties of Lasso-type estimators by performing extended simulations in a realistic panel data setting mimicking closely the behavior of the expected returns cross-section with a prediction target.

The paper is organized as follows. Section 2 provides a description of the relevant methodology. This section is followed by a description of the estimation objective and how it relates to a factor structure. Section 3 presents the simulation study. The data are briefly discussed in Section 4. The penultimate section covers the empirical work, including the return prediction and FC selection results. The final section concludes.

## 2. Methodology

This section introduces the notation and presents the underlying estimation methods and the statistics we use to evaluate the selection and prediction performance of the methods in our simulations and the empirical analysis.

### 2.1. Notation

Generally, if not explicitly otherwise stated, we follow the notation that $n$ refers to stock $n$ of $N_t$ total stocks and $t$ to period (i.e., month) $t$ of $T$ total periods. Moreover, factors are indexed by $c$ of $C$ total factors and belong to set $\mathcal{C}$, where each $c$ belongs to one of the following three groups or subsets: priced factors are denoted by $p$ (of a total $P$) and define the set $\mathcal{P}$; unpriced factors are defined by $u$ (of a total $U$, set $\mathcal{U}$), and spurious factors with respect to the return process are described by $s$ (of a total $S$, set $\mathcal{S}$). The total number of factors, $P + U + S = C$. A specification of each type of factor is outlined in more detail in Section 2.2. The indexing of FC is identical to that of the factors.

### 2.2. FC and the Return Generating Process

Generally, we assume a Rosenberg [32] and Daniel and Titman [33] type cross-sectional return structure. Covariances are determined based on a factor structure and expected returns mark a compensation for factor risk (default assumption). Following Daniel and Titman [33], we consider the following excess return generating process,

$$R^e_{n,t} = \beta'_{n,t-1} f_t + x'_{n,t-1} \delta + \eta_{n,t}, \quad t = 1, ...., T, \quad n = 1, ..., N \tag{1}$$

where $f_t$ defines the vector of factor returns of length $C$ and $\eta_{n,t}$ each stock idiosyncratic noise component, assumed to be normally distributed and orthogonal to the factors and other stocks' idiosyncratic components. $x'_{n,t} = [1 \ C_{n,t}]$, the vector of the corresponding $C$ FC and an intercept: $C_{n,t} = [size_{n,t}, bm_{n,t}, mom_{n,t}, ...]'$. Each factor, $f_t$, follows the dynamics,

$$f_{i,t} = \mu_i + \epsilon_{i,t}, \quad \text{with} \quad \epsilon_{i,t} \sim \mathcal{N}(0, \sigma_i^2) \quad i = 1, ..., C,$$

where $\mu_i$ defines the risk-premium of the $i$'th factor and $\epsilon_{i,t}$, $i = 1, \cdots, C$, the sequence of independent factors' innovations. Moreover, we assume a linear functional relation of the FC and the factor exposures,

$$\beta_{n,t} = g(x_{n,t}) = a_t + B x_{n,t} \tag{2}$$

with $B$ a $C \times C$ matrix of coefficients. Note that $a_t$ cancels, once we consider de-meaned cross-sectional returns. Naturally, exposures are time-varying. This is in line with the

empirical characteristics, as momentum or value exposures vary with the price level movements of each stock. The model 1 then becomes

$$
\begin{aligned}
R^e_{n,t} &= \beta'_{n,t-1}\mu + \beta'_{n,t-1}\epsilon_t + x'_{n,t-1}\delta + \eta_{n,t} \\
&= (Bx_{n,t-1})'\mu + (Bx_{n,t-1})'\epsilon_t + x'_{n,t-1}\delta + \eta_{n,t} \\
&= x'_{n,t-1}(B'\mu + \delta) + \widetilde{\eta_{n,t}}
\end{aligned}
$$

where $\widetilde{\eta_{n,t}}$ is the new zero mean innovation. As a consequence, the linear predictive dependence we aim to measure is of the form:

$$
E_{t-1}[R^e_{n,t}] = x'_{n,t-1}\gamma. \tag{3}
$$

To allow for different interpretations of the relationship between FC and expected returns from an asset pricing perspective we differentiate among three types of factors, namely priced, unpriced, and spurious factors:

- $P$ priced factors: $\mu_p \neq 0 \,\forall p \in \mathcal{P}$,
- $U$ unpriced factors: $\mu_u = 0 \,\forall u \in \mathcal{U}$,
- $S$ spurious factors: $\beta_{s,t} = 0$ and $\delta_s = 0, \forall s \in \mathcal{S}$.

Examples of priced factors are the market or value factor; for unpriced factors, sector factors; and for spurious factors, an independently created random time series. In a first model setting we consider risk-premia always coupled to the underlying risk-exposure, that is $\delta = 0$ and $\gamma_i = b_i\mu = 0 \,\forall i \in \mathcal{U}$, where $b_i = B_{.i}$ denotes the $i$-th column of $B$. As an example, the CAPM can be found in this asset pricing model interpretation by considering only one priced factor, the market factor and no unpriced or spurious factors.

Under a second asset pricing modeling interpretation, we consider a model where $\delta_i$ is not constrained to be equal to zero for the unpriced factors in (3). In case $\gamma_i = \delta_i \neq 0$ for some $i \in \mathcal{U}$, $\delta_i$ measures the sensitivity of FC to expected returns that do not directly compensate for factor risk. It imposes a non-zero covariance between FC ($x_{n,t-1}$) and some factors in case the FC are linked to the non-zero elements in $\delta$ as described in Daniel and Titman [33]. In this model setting, we might have zero-priced factors. In particular, this asset pricing model allows two stocks with an identical book-to-market ratio to have different risk exposures to a book-to-market value factor. Here the return compensation is associated with the book-to-market characteristic, i.e., mispricing, and not its risk sensitivity to the value dimension. The first asset pricing modeling framework rules this out. The second asset pricing model implies the presence of asymptotic arbitrage. Regardless of the interpretation, both models are estimated using (3).

*2.3. Methods*

We focus on three different linear models, which are defined as:

$$
\hat{\beta}_{\text{ols}} = \arg\min_{\beta}\left(\|\mathbf{Y} - \mathbf{X}\beta\|_2^2/n\right), \tag{4}
$$

$$
\hat{\beta}_{\text{Lasso}}(\lambda) = \arg\min_{\beta}\left(\|\mathbf{Y} - \mathbf{X}\beta\|_2^2/n + \lambda\|\beta\|_1\right), \tag{5}
$$

$$
\hat{\beta}_{\text{adapt}}(\lambda) = \arg\min_{\beta}\left(\|\mathbf{Y} - \mathbf{X}\beta\|_2^2/n + \lambda\sum_{j=1}^{p}\frac{|\beta_j|}{|\hat{\beta}_{\text{init},j}|}\right) \tag{6}
$$

where $Y \in \mathbb{R}$ and $X \in \mathbb{R}^p$ and the corresponding response vector $\mathbf{Y}_{n\times 1}$, the design matrix $\mathbf{X}_{n\times p}$, the parameter vector $\beta_{p\times 1}$. We slightly deviate in this subsection and denote the regression coefficient as $\beta$ (vs. $\gamma$). In all other sections we use the term $\beta$ exclusively as a measure of factor exposure, and $\gamma$ as the regression coefficient we aim to estimate. Moreover, throughout this work we treat $\mathbf{Y}$ and $\mathbf{X}$ as standardized matrices, with $\mu = 0$ and $\sigma = 1$, where the standardization is applied column by column. As defined in (1)

**Y** corresponds to the vector of excess returns, $R^e$, and **X** to the matrix of FC. Equation (4) defines the ordinary least squares (OLS) estimator, Equation (5) the Lasso estimator (Tibshirani [6]) and (6) the adaptive Lasso (Zou [7]). The Lasso and the adaptive Lasso differ in terms of the penalization term, which allows the weights to vary for each parameter. The assigned individual weights are inversely proportional to a first-stage $\beta$ estimate. Zou [7] suggests the use of the OLS estimator, $\hat{\beta}_{\text{ols}}$ as $\hat{\beta}_{\text{init}}$, unless collinearity is an issue. Bühlmann and Van De Geer [23] set $\hat{\beta}_{\text{init}} = \hat{\beta}_{\text{Lasso}}(\lambda)$. The use of the Lasso as a first-stage estimator is justified by the screening property of the Lasso, which still allows consistent variable selection of the adaptive Lasso at the second stage. We solely use the Lasso as a first stage estimator in (6) in this work. The penalty term $\lambda$ used in (5) and (6) is determined by cross-validation (CV) or classical selection criteria, typically five-fold or ten-fold CV, the Bayesian information criterion (BIC), or the Akaike information criterion (AIC). Bühlmann and Van De Geer [23] show that the optimal $\hat{\lambda}$ based on the BIC evaluation reads as follows:

$$\hat{\lambda}_{BIC} = \arg\min_{\lambda}\left( n\log\left(\frac{\|\mathbf{Y} - \hat{Y}_{\lambda}\|^2}{n}\right) + \log(n)\|\hat{\beta}_{\lambda}\|_0^0\right),$$

and accordingly the AIC,

$$\hat{\lambda}_{AIC} = \arg\min_{\lambda}\left( n\log\left(\frac{\|\mathbf{Y} - \hat{Y}_{\lambda}\|^2}{n}\right) + 2\|\hat{\beta}_{\lambda}\|_0^0\right).$$

Alternatively, the optimal $\lambda$ can be estimated by cross-validation. Here we randomly split the samples along time points and never within a given period. Assume we observe $T$ periods each containing $N$ stocks $S_{1,t}, S_{2,t},..., S_{N,t}$. Consequently, we can randomly select a training and testing set along the time index $t$. Hence, high cross-sectional correlations cannot cause biased estimates for the optimal $\lambda$.

The empirical set-up presented above makes shrinkage methods like the ones introduced above an attractive choice as they possess the ability to reduce the variance at the cost of slightly increasing the bias. First, as the variance increases in $p$, the ratio of $\frac{p}{T}$ can potentially be high, as we have 400+ presented factors in the literature and in the best case 50 years of monthly data ($T = 600$). Moreover, if some FC is available only for a shorter period of time, we can still perform the regression, as the Lasso methods are feasible even for the case where we have a truly high-dimensional problem ($p > T$), which imposes a constraint for classical OLS. Moreover, the noise component makes up unambiguously a significant proportion of the return process (even when assuming that the efficient market hypothesis is violated). Hence, the noise variance component has an important impact.

### 2.4. Data Sparsity

It is important to highlight the role played by the assumption of data sparsity connected to the use of the (adaptive) Lasso. Data sparsity is generally an untestable assumption and we consider it only a rough although reasonable approximation of reality. According to Zhang et al. [34] the concept of exact sparsity can be relaxed while still maintaining the same rate of convergence of the Lasso estimator to the true coefficients. They define that a model is sparse if most coefficients are small, in the sense that the sum of their absolute values is below a certain level. Under this general sparsity assumption, it is no longer sensible to select exactly the set of nonzero coefficients. Therefore, in cases where the exact selection consistency is unattainable or undesirable, the authors show that the Lasso is able to select the important variables with coefficients above a certain threshold determined by the controlled bias of the selected model. Thus, under this generalized sparsity concept, the (adaptive) Lasso is able to successfully discriminate between small and large coefficients and identify with high probability the most important firm characteristics; see also Bühlmann and Van De Geer [23] for a general review of the corresponding theory.

Moreover, given that our interest focuses primarily on the predictive ability of the competing approaches, the results discussed by Greenshtein et al. [35], Bickel et al. [36], and Sirimongkolkasem and Drikvandi [37] are reassuring: They highlight the fact that assuming sparsity as an approximation of the true design of the data does not generally significantly degrade the predictive accuracy of the models in high-dimensional settings with a large number of covariates. Greenshtein et al. [35] show that under various sparsity assumptions there is "asymptotically no harm" in considering a large number of covariates (many more than observations) for prediction purposes in a linear regression model under an $l_1$ constrained optimization. Bickel et al. [36] provide bounds on the $lp$ prediction loss, $1 \leq p \leq 2$, of the Lasso in a high dimensional linear regression in terms of the best possible (oracle) approximation under the sparsity constraint. Finally, by comparing different shrinkage approaches in a linear regression simulation setting, Sirimongkolkasem and Drikvandi [37] show that when important covariates are associated with correlated data, the $l_1$ and $l_2$ prediction performances of the Lasso improve for both sparse and non-sparse high dimensional settings and even sometimes outperform those of the Ridge regression. The predictive performance of the Lasso remains generally unaffected when the correlated covariates are associated with nuisance and less important variables. Given the previous evidence and the fact that the focus of the current study is set on identifying the most relevant methods and firm characteristics for predicting the cross-section of expected returns in a variable selection framework, we do not report results for alternative shrinkage techniques like the Ridge. Predictive performance results using Ridge are qualitatively similar to those presented for the lasso in Section 5.2.1 and are available from the authors upon request.

### 2.5. Selection and Prediction Evaluation

We apply pooled ordinary least squares as described in (4), where the t-values are based on the Driscoll and Kraay [38] robust standard errors. Next, we set a significance level for the OLS estimates to have a rule determining whether or not a coefficient can be seen as selected—we set the level to the literature standard of 5%. The impact of multiple-testing is gauged by considering t-value corrections as presented by Harvey et al. [1]. Specifically, we use the Bonferroni and Holm adjustment, which belongs to the class of family-wise error rates. Additionally, the study includes Benjamini, Hochberg and Yekutiel's (BHY) adjustment, a false discovery rate control, which we also consider; we refer to Harvey et al. [1] for a more complete description of the multiple-testing adjustments. In the case of the Lasso and the adaptive Lasso, the FC selection procedure is straightforward: all non-zero coefficient estimates are considered to be selected. Here we provide estimates for Lasso- and adaptive Lasso-based BIC, AIC and five-fold cross-validation (CV5) optimized regularization strength.

Following the variable selection, we calculate expected returns for each stock at each point in time. In this step, we evaluate two variants of each method. The first case drops all insignificant coefficients in the case of OLS and takes the relevant ones directly into consideration for the prediction. The second variant performs a post-variable selection OLS (PVSOLS).

The prediction quality is then measured based on a cross-sectional average, as proposed by Gu et al. [21]. More formally,

$$l_t = \frac{1}{N_t} \sum_{n=1}^{N_t} (R_{n,t}^e - \hat{R}_{n,t}^e)^2 \tag{7}$$

where $R_{n,t}^e$ and $\hat{R}_{n,t}^e$ denote the actual and the predicted excess returns over the risk-free rate, respectively. We report two metrics measuring the prediction performance, the simple time series average of $l_t$ and the model confidence set (MCS) as introduced by Hansen et al. [39].

Furthermore, we report the out-of-sample $\mathbb{R}^2_{OS}$ following Campbell and Thompson [40] defined as follows:

$$\mathbb{R}^2_{t,OS} = 1 - \frac{l_t}{\frac{1}{N_t}\sum_{n=1}^{N_t} R_{n,t}^{e\,2}} \quad (8)$$

## 3. Simulation Study

In order to analyze the suitability of the Lasso-type estimators in the previously described context, we propose a simulation of cross-sectional returns and FC. It is calibrated such that crucial properties of the cross-sectional return data are satisfied. Although the simulation setting is highly stylized and cannot be a perfect replication of the true underlying data-generating process (DGP), it is helpful for gaining insights into the method's model selection and predictive performance in finite samples under different distributional assumptions in our specific setting; see the literature review in Section 1 on what has already been proved theoretically.

### 3.1. Calibration

The calibration of the return-generating process is as follows:

- We set the number of priced and unpriced factors to 6 each. Even if it were theoretically assumed that unpriced factor risk should not exist, it could empirically still be present. Moreover, we assume that the factors are independent of one another and that the stock market factor explains the highest proportion of variance of all priced factors.
- Firm characteristics fall into one of the following three groups: Group 1 measures factor exposure to priced factors and group 2 to unpriced factors. Group 3 measures FCs which are independent of the return-generating process. FCs are potentially correlated across groups.
- The signal-to-noise ratio is relatively small, assuming a ratio implying a yearly $R^2$ of 5% (see the Appendix A for details on the transformation to monthly $R^2$s). This is in line with empirically documented $R^2$s in the case of the linear model; see, for example, Lewellen [41].
- The stock market factor follows a time-varying volatility process (implying heteroskedasticity for the individual stocks over time as well).
- The simulated return series do not possess any auto-correlation.
- The return-generating process follows (1) with $\epsilon_{c,t+1} \sim \mathcal{N}(0, \sigma_f^2) \,\forall c \in \mathcal{P} \cup \mathcal{S} \wedge c \neq 0$ and $\epsilon_{0,t+1} \sim \mathcal{N}(0, \sigma_{0,t}^2)$. FC, $c = 0$, represents the stock market return, which follows a latent volatility process $\sigma_{0,t}^2$ (see next item). $\epsilon_{c,t+1}$ is set to $0 \,\forall c \in \mathcal{S}$ (spurious factor). The elements of the vector of risk-premia, $\mu_c$, are drawn from $\sim \text{unif}(0.1, 3.5)\forall c \in \mathcal{P} \wedge c \neq 0$ and are set equal to zero $\forall c \in \mathcal{U} \cup \mathcal{S}$. We assume a stock market premium, $\mu_0$, of 5.5% per year. The risk-premia are drawn only once per case and are kept constant through each simulation. The market premium is the estimate of the Fama French market factor.
- $\sigma_{0,t}^2$, the stock market volatility, is estimated by using a GARCH(1,1) process, where the estimated $\hat{\sigma}_t$ are obtained by fitting a GARCH(1,1) model on daily observed US stock market returns. The GARCH(1,1) model captures a sufficient fraction of distribution properties observed in stock returns for our simulation study. Moreover, model performance seems reasonable compared to many less parsimonious approaches (see Hansen and Lunde [42]). Better volatility models exist, but are beyond the scope of this paper and not of crucial relevance.
- $\eta_{n,t}$ represents the idiosyncratic stock-specific component and is drawn from $\sim \mathcal{N}(0, \sigma_{idio}^2)$. Bekaert et al. [43] show that aggregated idiosyncratic volatility varies over time. Despite this empirical evidence, we choose a parsimonious approach to model idiosyncratic volatility. This is mainly motivated by the statistical properties our DGP already possesses.

- $x_{n,t}$ marks the vector of FC of stock $n$ at $t$ of length $C$. We simulate the characteristics $x_t$ from $\sim \mathcal{N}(0, \Sigma)$.
- The correlation matrix of FC, $\Sigma$, is obtained following the simulation approach of Hardin et al. [44] and is a crucial feature of our simulation. It is important, as many FC measure empirically similar variations. The base case refers to the constant correlation structure within groups (Algorithm 1) in Hardin et al. [44]. The $\Sigma$ is drawn only initially and kept constant in each specification. The empirical correlation structure described in Section 5.1 shows a handful of cases with pairwise correlations around 0.9 and many between 0.4 and 0.5. Our simulated correlation pairs reflect this, in order to investigate the impact of this difference. However, this study does not consider a correlation grouping of more than two FCs, or any other forms of more involved linear dependencies.

Finally, the collection of all $T$ periods of the simulations can then be stacked together in matrix $X$ and matrix $Y$. The true coefficient of interest is the vector of $\mu$, which is estimated as the vector of coefficients $\hat{\gamma}$.

The specification allows flexible simulations under different assumptions, analyzing the sensitivity of simulation parameters on the performance of the method. For this, we perform several different simulation specifications, each loosening one assumption separately. In general, the following default parameters are set:

- Number of simulations, 2000.
- Number of in-sample periods, 600, corresponding to 50 years of monthly data.
- Number of out-of-sample periods, 240, corresponding to 20 years of monthly data.
- Number of stocks, 4000, where 4000 corresponds to the average number of stocks used in the empirical part.
- Number of firm characteristics, 100—with $P = 6$, $U = 6$ and $S = 88$.
- $\sigma_f^2$ and $\sigma_{idio}^2$ result from the pre-specified level of $R^2$, where the noise variance is distributed as described in the Appendix A.
- The correlation matrix, $\Sigma$, is simulated such that we have one high (0.9) and one low (0.4) pairwise correlation between FC from each group. See Figure 1 for a visualization of one realization of the specified correlation matrix simulation.

*3.2. Sensitivity Analysis*

The behavior of the simulated DGP crucially depends on its calibration defined above. In order to investigate the sensitivity to these choices, we define the following cases:

**Case 1: Base case**
Default settings.

**Case 2: Small $T$**
Default settings, $T$ is set to 240.

**Case 3: Large $T$**
Default settings, $T$ is set to 4200 (and $N$ reduced to 800 to keep it computationally tractable). This specification requires a longer than available estimated GARCH(1,1) series; the missing $\sigma_{0,t}^2$ are simply simulated based on the GARCH(1,1) parameter estimates described in the Appendix A. (This case is not a realistic scenario for monthly data but is insightful for applications with higher frequencies.)

**Case 4: Expected returns: a function of FC instead of factor exposure**
Default settings. The premium of the stock market factor is set to zero. Instead, we attach the premium to the FC directly and impose a correlation of the factor exposure and the FC of 0.9. This is in line with our second asset pricing interpretation introduced in Section 2.2.

**Case 5: Small $N$**
Default settings, $N$ is set to 250.

Note that each simulation considers a balanced panel. As the actual data consist of an unbalanced panel, we adjust the data as described in Section 5.1. Moreover, the simulation ignores potential measurement errors in FC and assumes that they are measured

without errors. Empirically, the most common FC suffering from measurement error are, as mentioned above, market betas.

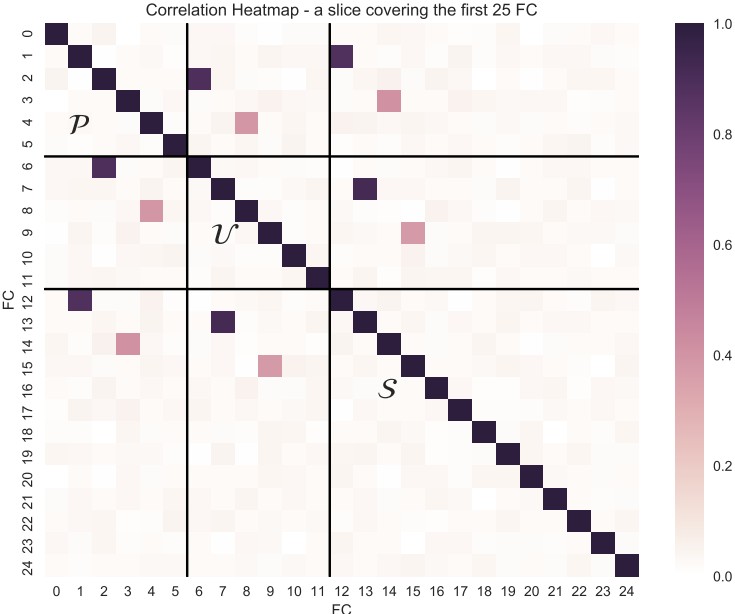

**Figure 1.** The figure displays a fraction of the default correlation matrix used for the simulations. It shows the correlation for the first 25 FC, where FC 0 to 5 refer to the set of FC with positive risk-premia, 6-11 belong to $\mathcal{U}$ and the rest to $\mathcal{S}$. For example, FC 1 (part of $\mathcal{P}$) and FC 12 (part of $\mathcal{S}$) are correlated with each other with a correlation of around 0.9. Note that the missing 75 FC are uninteresting insofar as their off-diagonal elements are close to zero.

*3.3. Performance Evaluation*

After each data simulation is performed, we report ten specifications covering all three methods and their respective choice variables presented above in the pooled panel framework. We collect the results of the OLS estimates, the Lasso and the adaptive Lasso. The evaluation considers two performance dimensions. First, we show the selection qualities of each method; second, we assess the simulated forecast accuracy, reflecting jointly the selection and parameter estimation qualities.

Knowing the true DGP of the simulation, we can then simply calculate the ratios of correctly classified coefficients, providing insights on type I and type II error behavior. Type II errors are calculated for FC 0–5—a failure to reject the null hypothesis of an unpriced factor FC given the factor is truly priced. Type I errors are measured for FC 6–100—rejecting the null hypothesis of an unpriced factor exposure given the FC is tied to an unpriced factor (FC 6–11) or the FC is independent of the returns (FC 12–100). The prediction evaluation follows Section 2.5 and is based on the 240 periods of simulated out-of-sample data.

3.3.1. Prediction

**Case 1: Default settings**

The prediction simulation results of case 1 are displayed in Figure 2. The BIC and CV5 Lasso specifications perform best considering the average p-value of the MCS; however, the differences are not large. The weakest among the methods is the standard POLS. Correcting the t-values for multiple testing yields meaningful improvements in this specification. The MSE ratios are more concentrated and only a few differences can be documented; the highest value achieves the Bonferroni and Holm t-value adjusted predictions.

**Case 2: Small *T***

Reducing the number of estimation periods increases the difference between the min and max p-value average. POLS remains the weakest prediction tool and the two Lasso-

type methods still perform best, although in the opposite order. Furthermore, the p-value corrected POLS-based predictions rank among the worst-performing methods. Consistent with the wider p-value difference, the MSE relative difference increases compared to the default case. Once more, adjusting the t-values for the variable selection realizes the highest MSE on average.

**Case 3: Large *T***

A larger *T* leads to convergence of the prediction results as the gap between the average p-value and the MSE narrows.

**Case 4: Direct linear dependence between FC and expected returns**

There is no notable difference in the order of the *p*-values; the two Lasso-type predictions yield the best results.

**Case 5: Small *N***

Compared to the previous four cases the Lasso CV5 methodology falls by some rank levels; however, the Lasso BIC remains on top. Overall, a lower number of stocks in the cross-section impacts the relative quality of POLS-based predictions negatively when compared to the default case.

**Figure 2.** The figure shows the simulated out-of-sample prediction evaluation for each method. Pooled OLS (POLS), Lasso AIC with post-variable selection OLS (L AIC PVOLS), pooled OLS with post-variable selection OLS (POLS PVOLS), Lasso CV5 with post-variable selection OLS (L CV5 PVOLS), adaptive Lasso CV5 (AL CV5), Lasso BIC with post-variable selection OLS (L BIC PVOLS), Lasso AIC (L AIC), adaptive Lasso BIC (AL BIC), adaptive Lasso CV5 with post-variable selection OLS (AL CV5 PVOLS), adaptive Lasso AIC (AL AIC), adaptive Lasso AIC with post-variable selection OLS (AL AIC PVOLS), pooled OLS with post-variable selection OLS and the BHY t-value adjustment(POLS PVOLS BHY), pooled OLS with post-variable selection OLS and the Bonferroni t-value adjustment (POLS PVOLS Bonf), pooled OLS with post-variable selection OLS and the Holm t-value adjustment(POLS PVOLS Holm), adaptive Lasso BIC with post-variable selection OLS (AL BIC PVOLS), Lasso CV5 (L CV5) and Lasso BIC (L BIC). The plot on the left side shows average MCS p-values over all 2000 simulation cases, where the p-value measures if the model is part of the MCS. The right figure illustrates the MSE values relative to the max of each case. The darker the color the better the performance relative to the competing methods.

### 3.3.2. Selection

**Case 1: Default settings**

The selection simulation results of case 1 are displayed in Table 1. The results reveal that there are distinct differences between the methods applied. It shows that for the

simulated stock market factor, OLS performs the worst, as it displays a type II error rate of around 50%. On the other hand, the Lasso and the adaptive Lasso methods show a far better performance with an error rate of around 0% and 15%, respectively, for the stock market factor (in the case of CV5). For the other, significantly less volatile factors carrying a positive risk-premium, the type II error rates are zero in the case of AIC and BIC-based selection. The exceptions are some POLS t-value adjusted estimates and all Lasso- and adaptive Lasso-selected FCs are based on CV5. The type I error cases have to be distinguished in two cases: First, for FC belonging to set $\mathcal{U}$ (unpriced factor FC) where OLS has slight advantages over the adaptive Lasso, only the CV5-based adaptive Lasso selections perform comparably. The Lasso reveals a poor performance with error rates mostly above 50%. The second case of type I errors comprises spurious FC. Correlations of the uninformative FC with any of the two other FC types prove once more to be a problem for the Lasso. For these cases, the more restrictive adaptive Lasso reveals a far better selection performance than the Lasso, as the error rate is zero for all cases. Moreover, the OLS type I error rate behaves as expected by varying around the 5% significance level.

**Case 2: Small *T***

Reducing the number of periods in the simulation reveals differences in the performance compared to the default simulation results, as Table 1 shows. First, type II error rates rise strongly for OLS estimates, whereas only a slight increase is observed for the adaptive Lasso and practically no changes are visible for the Lasso. The picture also changes when looking at type I error rates for FC in set $\mathcal{U}$, where we observe a jump in error ratios for the adaptive Lasso.

**Case 3: Large *T***

As T grows larger, the error ratios decline as expected. The only remarkable exception is found for the Lasso, where once more the correlated FC remains prone to false inference. A strong indication that the neighborhood stability condition is likely to be violated in cases of higher correlations is that the error rate of FC 6 is 99% and that of FC 12 reaches 84%. Moreover, higher error ratios are also observed for cases with weaker correlations of around 0.5; see FC cases 7–9 and 12–15.

**Case 4: Direct linear dependence between FC and expected returns**

This specification yields a performance improvement for all methods, most apparent for OLS, where the classification error comes down from 33% to 23% compared to the default assumption.

**Case 5: Small *N***

We observe some higher type II errors for one FC in the case of adjusted POLS selection and generally lower type I errors for the Lasso-type methods.

Finally, we briefly summarize the simulation results. We show that the adaptive Lasso is superior to OLS when type II errors are a concern. A Lasso-based selection reveals for this case only negligible advantages over the adaptive Lasso. The picture changes if we want to minimize type I error behavior. Here we have to differentiate between two distinct scenarios. First, whenever we encounter an entirely uninformative independent variable, we show that the adaptive Lasso performs best. Second, in case we have a relation of the independent variable with an unpriced risk factor of the dependent variable, an OLS approach achieves the best results. We note that correlations are a crucial driver behind these results, where, in particular, the Lasso presents problems reaching reasonable type I error ratios when confronted with higher correlations ($\approx$0.9). Moreover, altering the optimal $\lambda$ selection mechanism impacts the results importantly. BIC is favorable over AIC in the specifications under consideration. BIC reduces type I errors without suffering from an increase in type II misclassifications. BIC vs. CV exposes a tradeoff between type I and type II. Assigning equal weight to both error types, BIC-based estimation is the preferable tuning method. Additional robustness checks can be found in Appendix A.2.

**Table 1. Simulation Results:** The table provides an overview of type II (for priced factor FC, FC 0–5) and type I error (for unpriced factor FC, 6-11, and spurious FC, 12–99) ratio behavior in percentage points for the specified simulation cases. All details of the cases can be found in Section 3.2. FC cases 12–15 are interesting insofar as they are spurious FC with high correlations to priced and unpriced FC. A white space represents a zero value.

| Case | Method | $\mathcal{P}$ — Type II | | | | | | $\mathcal{U}$ — Type I | | | | | | $\mathcal{S}$ — Type I | | | | Summary FC 16-99 | | | |
|---|---|---|---|---|---|---|---|---|---|---|---|---|---|---|---|---|---|---|---|---|---|
| | | 0 | 1 | 2 | 3 | 4 | 5 | 6 | 7 | 8 | 9 | 10 | 11 | 12 | 13 | 14 | 15 | mean | std | min | max |
| 1 | POLS | 0.50 | | | | | | 0.05 | 0.04 | 0.06 | 0.06 | 0.05 | 0.05 | 0.05 | 0.05 | 0.05 | 0.04 | 0.05 | 0.01 | 0.04 | 0.06 |
| | POLS BY | 0.88 | 0.06 | 0.04 | | | 0.02 | | | | | | | | | | | | | | |
| | POLS Holm | 0.89 | 0.06 | 0.04 | | | 0.02 | | | | | | | | | | | | | | |
| | POLS Bonf | 0.90 | 0.06 | 0.04 | | | 0.02 | | | | | | | | | | | | | | |
| | AL BIC | 0.02 | | | | | | 0.01 | 0.11 | 0.11 | 0.12 | 0.12 | 0.13 | | | | | | | | |
| | AL AIC | 0.02 | | | | | | 0.01 | 0.13 | 0.14 | 0.16 | 0.15 | 0.16 | | | | | | | | |
| | AL CV5 | 0.14 | 0.10 | 0.11 | 0.09 | 0.08 | 0.10 | | 0.05 | 0.05 | 0.06 | 0.05 | 0.06 | | | | | | | | |
| | L BIC | 0.01 | | | | | | 0.40 | 0.60 | 0.63 | 0.65 | 0.65 | 0.63 | 0.28 | 0.18 | 0.05 | 0.02 | 0.01 | | | 0.01 |
| | L AIC | | | | | | | 0.52 | 0.78 | 0.83 | 0.84 | 0.85 | 0.83 | 0.42 | 0.35 | 0.29 | 0.26 | 0.25 | 0.01 | 0.21 | 0.27 |
| | L CV5 | 0.09 | 0.06 | 0.06 | 0.06 | 0.06 | 0.06 | 0.37 | 0.51 | 0.53 | 0.55 | 0.54 | 0.55 | 0.25 | 0.17 | 0.07 | 0.04 | 0.03 | | 0.02 | 0.04 |
| 2 | POLS | 0.52 | 0.04 | 0.17 | 0.12 | 0.05 | | 0.06 | 0.05 | 0.06 | 0.06 | 0.06 | 0.06 | 0.05 | 0.05 | 0.05 | 0.06 | 0.06 | | 0.04 | 0.07 |
| | POLS BY | 0.90 | 0.36 | 0.65 | 0.58 | 0.41 | 0.03 | | | | | | | | | | | | | | |
| | POLS Holm | 0.89 | 0.32 | 0.61 | 0.54 | 0.37 | 0.02 | | | | | | | | | | | | | | |
| | POLS Bonf | 0.89 | 0.33 | 0.62 | 0.54 | 0.37 | 0.02 | | | | | | | | | | | | | | |
| | AL BIC | 0.03 | | 0.07 | 0.01 | | | 0.10 | 0.36 | 0.32 | 0.38 | 0.37 | 0.37 | | | | | | | | |
| | AL AIC | 0.02 | | 0.05 | 0.01 | | | 0.11 | 0.39 | 0.37 | 0.43 | 0.41 | 0.41 | | | | | | | | |
| | AL CV5 | 0.15 | 0.11 | 0.25 | 0.15 | 0.12 | 0.08 | 0.05 | 0.15 | 0.13 | 0.16 | 0.15 | 0.15 | | | | | | | | |
| | L BIC | 0.01 | | 0.02 | | | | 0.47 | 0.74 | 0.74 | 0.76 | 0.79 | 0.78 | 0.29 | 0.23 | 0.05 | 0.04 | 0.01 | | 0.01 | 0.02 |
| | L AIC | | | 0.01 | | | | 0.63 | 0.85 | 0.89 | 0.90 | 0.90 | 0.90 | 0.39 | 0.37 | 0.28 | 0.27 | 0.26 | 0.01 | 0.23 | 0.29 |
| | L CV5 | 0.10 | 0.07 | 0.11 | 0.08 | 0.07 | 0.06 | 0.38 | 0.51 | 0.49 | 0.52 | 0.53 | 0.52 | 0.17 | 0.12 | 0.02 | 0.02 | 0.01 | | 0.01 | 0.02 |
| 3 | POLS | | | | | | | 0.06 | 0.05 | 0.06 | 0.05 | 0.05 | 0.05 | 0.06 | 0.05 | 0.04 | 0.05 | 0.05 | 0.01 | 0.04 | 0.06 |
| | POLS BY | 0.04 | | | | | | | | | | | | | | | | | | | |
| | POLS Holm | 0.04 | | | | | | | | | | | | | | | | | | | |
| | POLS Bonf | 0.04 | | | | | | | | | | | | | | | | | | | |
| | AL BIC | | | | | | | | | | | | | | | | | | | | |
| | AL AIC | | | | | | | | 0.01 | 0.01 | | | | | | | | | | | |
| | AL CV5 | | | | | | | | | | | | | | | | | | | | |
| | L BIC | | | | | | | 0.37 | 0.37 | 0.38 | 0.40 | 0.40 | 0.43 | 0.26 | 0.10 | 0.03 | 0.02 | 0.01 | | | 0.01 |
| | L AIC | | | | | | | 0.46 | 0.62 | 0.70 | 0.71 | 0.71 | 0.74 | 0.39 | 0.30 | 0.23 | 0.24 | 0.22 | 0.01 | 0.20 | 0.24 |
| | L CV5 | | | | | | | 0.42 | 0.49 | 0.52 | 0.56 | 0.54 | 0.57 | 0.33 | 0.19 | 0.11 | 0.10 | 0.06 | 0.01 | 0.05 | 0.08 |

**Table 1.** *Cont.*

| Case | Method | $\mathcal{P}$ — Type II | | | | | | $\mathcal{U}$ — Type I | | | | | | $\mathcal{S}$ — Type I | | | | Summary FC 16-99 | | | |
|---|---|---|---|---|---|---|---|---|---|---|---|---|---|---|---|---|---|---|---|---|---|
| | | 0 | 1 | 2 | 3 | 4 | 5 | 6 | 7 | 8 | 9 | 10 | 11 | 12 | 13 | 14 | 15 | mean | std | min | max |
| 4 | POLS | 0.34 | | | | | | 0.06 | 0.05 | 0.06 | 0.06 | 0.05 | 0.06 | 0.06 | 0.06 | 0.05 | 0.05 | 0.05 | | 0.04 | 0.06 |
| | POLS BY | 0.79 | | | | | | | | | | | | | | | | | | | |
| | POLS Holm | 0.80 | | | | | | | | | | | | | | | | | | | |
| | POLS Bonf | 0.81 | | | | | | | | | | | | | | | | | | | |
| | AL BIC | 0.01 | | | | | | 0.02 | 0.21 | 0.19 | 0.24 | 0.22 | 0.24 | | | | | | | | |
| | AL AIC | 0.01 | | | | | | 0.02 | 0.24 | 0.23 | 0.27 | 0.26 | 0.28 | | | | | | | | |
| | AL CV5 | 0.04 | | | | | | 0.10 | 0.09 | 0.11 | 0.10 | 0.11 | | | | | | | | | |
| | L BIC | | | | | | | 0.43 | 0.70 | 0.71 | 0.74 | 0.75 | 0.74 | 0.31 | 0.22 | 0.04 | 0.03 | 0.01 | | 0.01 | 0.01 |
| | L AIC | | | | | | | 0.58 | 0.83 | 0.87 | 0.87 | 0.89 | 0.90 | 0.42 | 0.37 | 0.28 | 0.29 | 0.26 | 0.01 | 0.23 | 0.28 |
| | L CV5 | 0.02 | | | | | | 0.42 | 0.58 | 0.58 | 0.62 | 0.62 | 0.62 | 0.25 | 0.16 | 0.04 | 0.04 | 0.02 | | 0.02 | 0.03 |
| 5 | POLS | 0.44 | | 0.02 | | | | 0.05 | 0.06 | 0.05 | 0.05 | 0.06 | 0.05 | 0.05 | 0.06 | 0.05 | 0.05 | 0.05 | | 0.04 | 0.06 |
| | POLS BY | 0.88 | 0.03 | 0.27 | 0.05 | | | | | | | | | | | | | | | | |
| | POLS Holm | 0.88 | 0.03 | 0.26 | 0.05 | | | | | | | | | | | | | | | | |
| | POLS Bonf | 0.89 | 0.03 | 0.27 | 0.05 | | | | | | | | | | | | | | | | |
| | AL BIC | 0.03 | | 0.01 | | | | 0.03 | 0.06 | 0.07 | 0.08 | 0.09 | 0.07 | 0.01 | 0.01 | | | | | | |
| | AL AIC | 0.02 | | | | | | 0.08 | 0.15 | 0.16 | 0.20 | 0.19 | 0.17 | 0.02 | 0.04 | 0.01 | 0.01 | | | | 0.01 |
| | AL CV5 | 0.07 | 0.01 | 0.03 | 0.02 | 0.01 | 0.01 | 0.04 | 0.06 | 0.07 | 0.07 | 0.08 | 0.08 | 0.01 | 0.01 | | | | | | |
| | L BIC | 0.02 | | | | | | 0.33 | 0.16 | 0.21 | 0.20 | 0.19 | 0.18 | 0.27 | 0.06 | 0.04 | 0.02 | 0.01 | | | 0.01 |
| | L AIC | 0.01 | | | | | | 0.42 | 0.44 | 0.52 | 0.54 | 0.55 | 0.55 | 0.38 | 0.25 | 0.23 | 0.20 | 0.20 | 0.01 | 0.18 | 0.22 |
| | L CV5 | 0.04 | 0.01 | 0.01 | 0.01 | 0.01 | 0.01 | 0.39 | 0.34 | 0.39 | 0.40 | 0.41 | 0.42 | 0.34 | 0.17 | 0.15 | 0.10 | 0.09 | 0.01 | 0.08 | 0.11 |

## 4. Data

Our objective is to preserve consistency as much as possible. Therefore the selection, data preparation and the notation of the description of firm characteristics generally follow the approach of Green et al. [3]. The FC data are implemented independently of Green et al. [3]. Our sample period ranges from 1974 to 2020. As in most studies, the analysis considers only CRSP stocks with share codes 10 and 11 which are traded either at NYSE, AMEX or NASDAQ; for an example, see Fama and French [4]. Furthermore, we exclude stocks with missing market capitalization data and/or where book values are unavailable. Compustat data are aligned with a standard lag of six months of the fiscal year end date. For example, the data of a firm with fiscal year end date 12/31 are aligned with data 06/30, predicting monthly returns from 6/30 to 7/31. CRSP-based stock/firm characteristics, such as idiosyncratic volatility, beta, maximum return or six-months momentum are used as of the most recent month end. For example, for the return prediction from 6/30 to 7/31, the max daily return from the period 5/31-6/30 is used. Additionally, following Green et al. [3] some selected Compustat accounting data are set to zero if not available; see the Appendix A for details. In processing larger amounts of data, correcting extreme and often implausible values is mostly unavoidable. Correcting these values on a discretionary basis is not feasible; hence, winsorizing the data is a useful strategy to reduce the problem. Therefore, each FC is winsorized at the 1% and 99% percentile at each point in time. Binary FC like *divi*, *divo*, *rd* and *ipo* are excluded from the winsorizing procedure. In the next step, missing data are replaced by the mean of the winsorized data at each point in time. Only then can the z-score standardization be applied at each calendar point. We do winsorize the return observations at the 5% and 95% percentile at each point to reduce the weight of outliers in the least-squares setting; therefore, no observations are excluded because of implausible returns. Moreover, returns are only de-meaned for each period and not corrected by the standard deviations. Finally, the data can be stacked and the pooled regressions applied, as each independent variable has mean zero and variance one given by the property of combining z-scores. Note that this is necessary as the Lasso requires a normalized design matrix as input, as described above.

However, differences in the selection of FC are unavoidable. This study employs only FCs which are not dependent on Compustat quarterly and IBES data. A detailed description of each FC included in the empirical part of this study can be found in the Appendix A. Moreover, the $\beta$ estimates are obtained by regressing rolling weekly stock returns on the market excess returns. The literature often employs an alternative procedure whereby stocks are ranked and sorted into portfolios according to their individual market beta; see, for example, Fama and French [4]. The betas assigned to each stock for estimating the equity market premia are obtained by using the betas of the corresponding portfolios. Using portfolio beta estimates instead of individual stock betas has been applied to reduce potential errors-in-variable issues in the second stage regression. However, Ang et al. [45] cast doubt on whether portfolio betas are optimal due to the loss of dispersion in individual betas. More details about the specific CRSP and Compustat data and the corresponding data alignment process can be found in the Appendix A. The returns used in the prediction regression are the CRSP returns (*RET*) adjusted by the provided CRSP delisting return (*DLRET*). Additionally and for verification purposes, we benchmark our data for selected FC with the FC portfolio returns provided by Kenneth French's Data Library. We find satisfying $R^2$s, reaching values from 0.99> to about 0.9 for cases where the FC definition of the benchmark data slightly deviates from the one presented in Green et al. [3]. Furthermore, we follow Fama and French [46] for the size classification definition, where large-cap stocks are the 1000 stocks with the highest market capitalization, mid-cap stocks rank 1001–2000 and small comprise all stocks with rank >2000. Finally, our industry-adjusted variables always use the 48 sectors downloaded from Kenneth French's Data Library, as the SIC 2 classification is empirically too granular since in many instances the sector group is defined by a single stock.

## 5. Empirical Results

The first subsection explains the details of how we construct the required normalized matrix *X* of the unbalanced panel of FC and returns. This subsection is followed by the empirical analysis of the predictability of cross-sectional stock returns. The third subsection covers the discussion of the selected FC.

### 5.1. Estimation Set-Up

As described above, the approach estimates the coefficients based on a pooled panel set-up. However, simply stacking the data causes two problems.

First, since we are interested in cross-sectional differences, we need to normalize each FC at each point in time to preserve the cross-sectional information. To illustrate the problem, one can think about the book-to-market ratio of single stocks, which certainly fluctuates partially based on market-wide price movements through time; standardizing along the entire panel would then implicitly change the order as time and cross-sectional information get mixed up.

Another issue that needs to be addressed is the unbalanced panel structure, as it implicitly causes the weights of each period in the regression to vary. Assuming there is no correlation between the returns and the number of stocks, we could ignore this issue, but empirically this is not the case; for example, we see that prior to the stock market peak at the beginning of the 2000s we have a much higher number of stocks with unknown return dependence. Therefore, we suggest adjusting the number of stocks in each period to the mean number of stocks per time point. This can be achieved by simply randomly drawing stocks with replacements at each point in time until we have filled the desired sample size.

Figure 3 presents the correlation structure of the FC. It shows overall only five cases with absolute correlation coefficients greater than 0.9. Even though we cannot achieve precisely the same correlation structure in our simulation, we have considered cases with correlations of around 0.9 and hence capture this feature observed in the data in our simulation as well. As we show in the simulation study in Section 3, correlations around 0.9 cause no selection issues for the adaptive Lasso; only a Lasso-based selection appears prone to misclassification. However, we want to avoid including almost identical FC. Hence, before regressing the returns on the full set of FC included in our dataset, we screen the correlations for cases with an absolute correlation greater than 0.95. In such cases, we eliminate the more recently published FC of the affected pair from our analysis. Finally, not all FC are included in our FC analysis due to data problems; we drop: cfp_ia, roic, pchemp_ia, and ipo.

### 5.2. Predicting the Cross-Section of Returns

The prediction results presented in this subsection analyze the out-of-sample return predictions from 1992 to 2020. We run monthly rolling and expanding window regressions to form predictions for the upcoming month. The expanding window regression is initially fit with 15 years of observations. The rolling window specification includes three different windows, with 10, 15 and 20 years of data. Moreover, we form five different data groups classified by market capitalization: all, including all stocks available; large, consisting of the highest 1000 ranked stocks; mid, the stocks ranked between 1001 and 2000; large plus mid, the top 2000; small, considering all stocks below a market cap rank 2000. Hence, we look in total at 20 different data groupings. Note that, the results in this section represent aggregated numbers, as we show averages of the four different data estimation windows defined above.

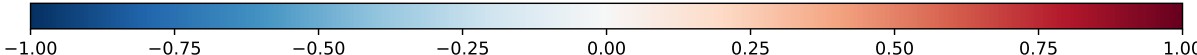

**Figure 3. Empirical correlation matrix** (best seen in color): The figure exhibits the correlation of all stocks and FC included in our analysis for the years June 1974 until December 2020. We use the normalized, winsorized and pooled FC data (as used in the full sample regression) to calculate the correlation coefficients. The figure shows five extreme correlation pairs (>0.9). For example, the highest absolute correlation is measured for *beta* and *beta_sq* with a coefficient slightly less than 0.95.

5.2.1. Performance Evaluation

Table 2 and Figure 4 present the empirical out-of-sample prediction evaluation. It reveals that the ability to forecast cross-sectional returns varies along the size dimension. Large-cap stocks are not predictable compared to a naïve benchmark for the full sample period, whereas small and micro-cap stocks are highly predictable. When focusing on $R^2$ statistics we do not find any meaningful differences along the time dimension.

Separating the predictions into a different sizes and time buckets allows for a more granular perspective. Figure 4 displays the 12-month aggregated rolling $R^2$ for each size

group. The prediction quality of the sample including all stocks shows relatively stable predictability, with a period of poorer forecasts between 2003 and 2008. However, small and micro-cap stock returns are predictable until the end of our sample. The MCS p-value for the naïve zero-return forecast is consistently below the 5% significance level. The question of whether these predictable returns are exploitable by investors remains unanswered here. It might reflect the fact that risk compensation or existing trading frictions simply do not allow prices to reflect all information available at the time. On the other hand, large-cap stock returns are much harder to predict during the entire sample, with the exception of a short period in the early 2000s. The prediction results highlight the importance of evaluating the quality of return predictions conditioned on size corroborating the empirical evidence shown in the previous literature. Splitting the sample into a pre- and post-2004 period (not reported) yields almost the same results for samples when assessing the predictability of cross-sectional returns compared to the full period analysis.

Comparing different linear estimators reveals that the empirical results of *all* stocks are consistent with the simulation findings. Based on the MSE and $R^2$ metric, the Lasso specifications perform best followed by the adaptive Lasso, whereas POLS-based methods show the weakest performance. Moreover, the five-fold cross-validation reaches the best prediction results in the case of Lasso and adaptive Lasso-based predictions. However, the differences are not statistically significant, as we can only reject the Lasso AIC case as not being part of the MCS at a 10% confidence level.

The predictability pattern changes if we consider only *large* cap stocks, as all out-of-sample $R^2$s turn negative. Overall, large-cap stocks are not predictable with the selected linear methods measured over the full sample between 1992 and 2020 as the zero return forecast achieves the lowest MSE. The prediction quality slightly improves when considering mid-cap stocks only, with two specifications achieving slightly positive $R^2$s. Doubling the cross-sectional sample size by combining large and mid-cap stocks does not improve the prediction quality in a statistical or economically meaningful way. The small-cap subset shows that small and micro-cap stocks are highly predictable as the zero return prediction benchmark is statistically rejected at a 1% level and not part of the MCS. Lasso-type predictions perform best in this case. Furthermore, the size sub-sample analysis underscores the importance of conditioning on the size as the existing predictability shown for all stocks is mostly driven by a small fraction of the market which only accounts for a negligible share of the US market capitalization. This result echoes the findings of Hou et al. [13], who show that small and micro-cap stocks contribute disproportionately to return characteristics of many published anomalies.

Furthermore, we can see that there are distinct differences in the number of selected FC. The Lasso specifications contain the largest number of FC; on average about 14–42 FC are selected to form expected returns. The more conservative adaptive Lasso selects around 5–28 FC depending on the tuning method. The POLS-based forecasts use a lower dimensional model, as it includes on average between 1 and 18 FC. Counting the pure number of FC can be misleading, as many coefficients might be close to zero and hence the effective dimensionality could be more similar between the methods. If we compare the absolute sum of all coefficients between the different regressions, we still see meaningful differences, however at a different order of magnitude than purely counting active variables.

**Table 2. Out-of-Sample Forecast Evaluation:** The MSE and $R^2$s are calculated according to (7)–(8). The MCS indicates the *p*-value of being part of the set that includes the best model. The monthly $R^2$s are expressed in percentage points. The MSE and $\sum$ abs(coef) column values are scaled by a factor of $10^3$ and $10^1$, respectively. The columns "Median #" and "Mean #" show the time-series median and mean of the number of active FC.

| Size Sample | Estimator | MSE | MCS | $R^2$ | Median # | Mean # | $\sum$ abs(coef) |
|---|---|---|---|---|---|---|---|
| All | AL AIC | 142.48 | 0.39 | 0.60 | 26.6 | 25.0 | 3.7 |
| | AL BIC | 142.49 | 0.21 | 0.60 | 21.5 | 20.8 | 3.2 |
| | AL CV5 | 142.48 | 0.52 | 0.60 | 16.8 | 15.9 | 2.7 |
| | L AIC | 142.50 | 0.05 | 0.59 | 41.8 | 42.0 | 4.6 |
| | L BIC | 142.47 | 0.69 | 0.61 | 31.8 | 31.9 | 3.4 |
| | L CV5 | 142.47 | 0.91 | 0.62 | 29.8 | 30.5 | 3.1 |
| | POLS PVSOLS | 142.49 | 0.49 | 0.58 | 18.0 | 17.8 | 3.3 |
| | POLS PVSOLS Holm | 142.52 | 0.50 | 0.54 | 6.5 | 6.5 | 1.8 |
| | Zero | 143.04 | 0.01 | 0.00 | 0.0 | 0.0 | 0.0 |
| Large | AL AIC | 73.48 | 0.17 | −0.13 | 21.9 | 20.5 | 2.1 |
| | AL BIC | 73.47 | 0.33 | −0.11 | 13.2 | 12.6 | 1.5 |
| | AL CV5 | 73.41 | 0.83 | −0.03 | 4.0 | 5.1 | 0.7 |
| | L AIC | 73.49 | 0.17 | −0.17 | 39.2 | 39.5 | 3.0 |
| | L BIC | 73.44 | 0.72 | −0.07 | 21.2 | 20.4 | 1.4 |
| | L CV5 | 73.45 | 0.48 | −0.06 | 11.1 | 14.0 | 0.9 |
| | POLS PVSOLS | 73.46 | 0.43 | −0.13 | 12.1 | 11.5 | 1.5 |
| | POLS PVSOLS Holm | 73.40 | 0.82 | −0.01 | 1.2 | 1.5 | 0.3 |
| | Zero | 73.39 | 0.89 | 0.00 | 0.0 | 0.0 | 0.0 |
| Large + mid | AL AIC | 97.60 | 0.60 | −0.05 | 23.2 | 22.3 | 2.4 |
| | AL BIC | 97.59 | 0.58 | −0.04 | 15.5 | 15.3 | 1.8 |
| | AL CV5 | 97.58 | 0.72 | −0.02 | 5.5 | 7.3 | 1.0 |
| | L AIC | 97.61 | 0.25 | −0.08 | 42.1 | 41.4 | 3.5 |
| | L BIC | 97.58 | 0.74 | −0.03 | 27.0 | 26.2 | 1.9 |
| | L CV5 | 97.58 | 0.73 | −0.02 | 17.4 | 18.0 | 1.3 |
| | POLS PVSOLS | 97.58 | 0.73 | −0.05 | 14.2 | 14.2 | 2.1 |
| | POLS PVSOLS Holm | 97.53 | 0.96 | 0.00 | 3.2 | 3.4 | 0.6 |
| | Zero | 97.52 | 0.84 | 0.00 | 0.0 | 0.0 | 0.0 |
| Mid | AL AIC | 118.75 | 0.16 | −0.04 | 24.0 | 23.4 | 2.7 |
| | AL BIC | 118.73 | 0.16 | −0.01 | 10.0 | 10.3 | 1.4 |
| | AL CV5 | 118.71 | 0.43 | −0.01 | 7.0 | 8.1 | 1.1 |
| | L AIC | 118.76 | 0.17 | −0.06 | 39.8 | 39.3 | 3.5 |
| | L BIC | 118.68 | 0.83 | 0.03 | 15.0 | 15.1 | 1.2 |
| | L CV5 | 118.70 | 0.45 | 0.00 | 15.8 | 17.2 | 1.4 |
| | POLS PVSOLS | 118.74 | 0.29 | −0.05 | 13.1 | 13.5 | 2.3 |
| | POLS PVSOLS Holm | 118.67 | 0.87 | −0.01 | 3.2 | 3.4 | 0.8 |
| | Zero | 118.65 | 0.86 | 0.00 | 0.0 | 0.0 | 0.0 |
| Small | AL AIC | 187.66 | 0.45 | 0.76 | 29.9 | 28.1 | 4.6 |
| | AL BIC | 187.67 | 0.33 | 0.76 | 19.0 | 18.5 | 3.5 |
| | AL CV5 | 187.68 | 0.31 | 0.75 | 19.2 | 18.3 | 3.4 |
| | L AIC | 187.67 | 0.09 | 0.75 | 41.4 | 41.1 | 5.3 |
| | L BIC | 187.64 | 0.83 | 0.77 | 26.0 | 25.0 | 3.3 |
| | L CV5 | 187.64 | 0.68 | 0.77 | 30.0 | 29.6 | 3.6 |
| | POLS PVSOLS | 187.69 | 0.35 | 0.73 | 17.8 | 16.7 | 4.0 |
| | POLS PVSOLS Holm | 187.81 | 0.16 | 0.61 | 5.8 | 5.9 | 2.2 |
| | Zero | 188.63 | 0.00 | 0.00 | 0.0 | 0.0 | 0.0 |

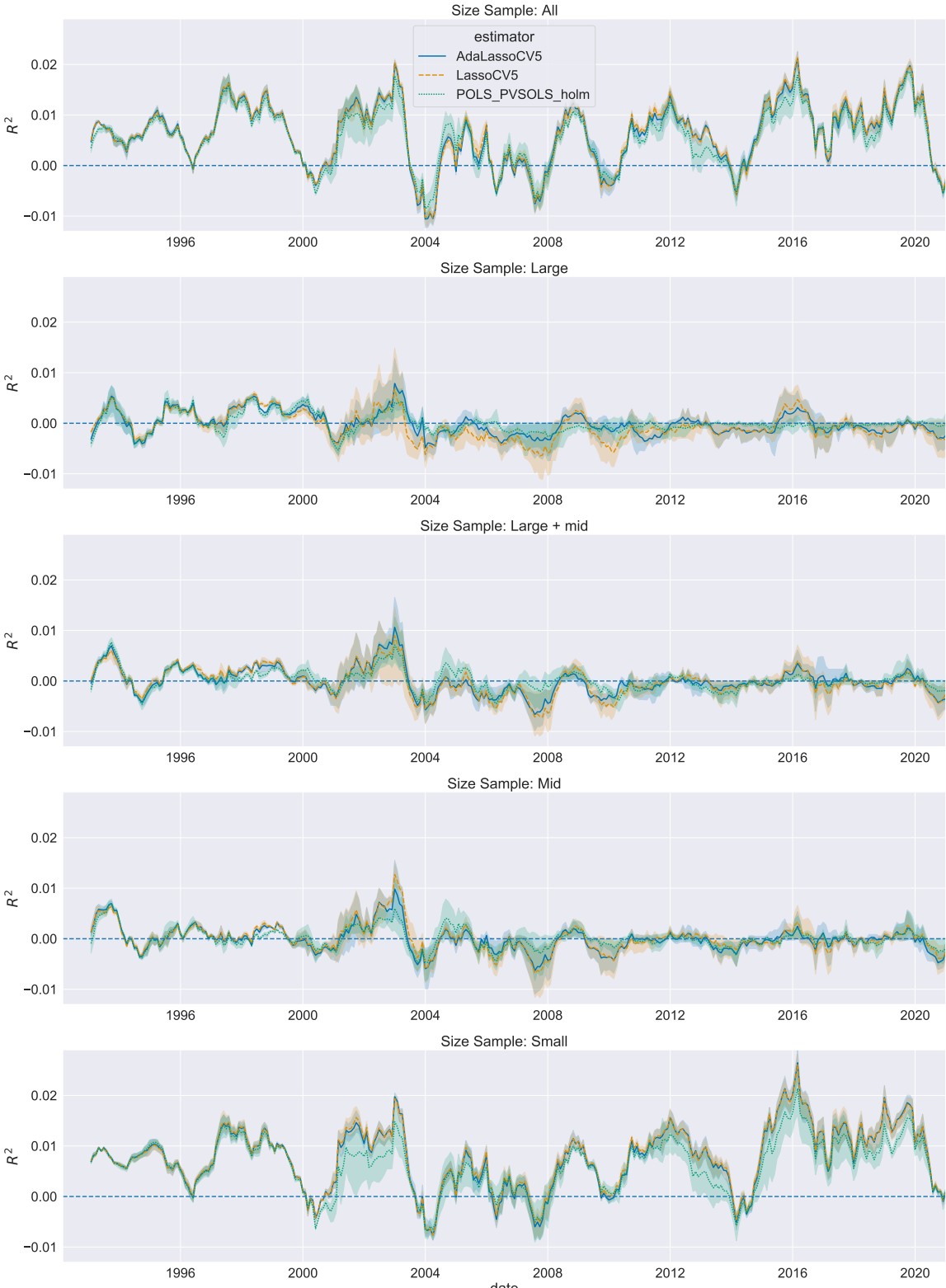

**Figure 4. 12-month rolling** $R^2$ (best seen in color): The figure shows the 12-month rolling $R^2$ as defined in (8) of each of the five different size groups. We aggregate the four different estimation window types per method and display the central tendency with confidence bounds.

### 5.3. Shrinking the Zoo of Firm Characteristics for Prediction

The analysis considers the years from 1974 to 2020 and primarily emphasizes the selection regression including all stocks. These findings are shown in Table 3. Furthermore, this table shows the regression results of conditioning on large, mid and small-cap-sized stocks. It displays the coefficient estimates of all FC determined by the Lasso, adaptive Lasso, and POLS. Given the results discussed in the previous section in terms of predictive accuracy, we focus on the lasso procedures optimized using five-fold cross-validation. The discussion in this section stresses mostly the details of the FC selection for the adaptive Lasso and the differences from the alternative selection procedures. As we showed in the simulations the adaptive lasso with five-fold cross-validation is able to reduce significantly the number of false positives. As a consequence, the firm characteristics estimated to belong to the active set by the adaptive lasso are quite reliable and give a clear indication of which characteristics carry relevant information for prediction.

Once more, results reflect the findings of Hou et al. [13], who document the impact of micro-cap stocks. It is also in line with the work of Green et al. [3], as they emphasize a value-weighted selection exercise. We are interested in the full sample analysis, i.e., the results of the single pooled regression applied at once over all periods to obtain the set of selected FC. Note that we denote the sign of the selected FC in brackets behind each FC when mentioned in the text for the first time and also provide a brief description of the respective FC but for the following subsection only. For the other subsections, we refer to Appendix A.

#### 5.3.1. FC Selection including All Stocks

Considering all stocks for the selection analysis, the first block of columns in Table 3 embodies the FC selection results for the adaptive Lasso, POLS and Lasso-based on all sample periods. Note that we drop *beta_sq* prior to the selection regression for all samples, due to its high correlation with *beta* within the large and mid-cap stocks. It is not surprising that *beta* and *beta_sq* are more highly correlated for large and mid-cap stocks, as the beta of these stocks tends to be more centered around one, causing any quadratic transformation to be more correlated compared to a more dispersed *beta* measured among small-caps. We find that dimensionality varies starkly between the methods. The adaptive Lasso selects 21, the Lasso 47, POLS with unadjusted t-values 23 and the DFDR adjusted POLS inference 13 FC. The adaptive Lasso selects four out of five FC associated with the Fama and French [9] five-factor model and observes consistency with respect to the sign of the coefficients. Specifically, we identify *beta*(+)—market; *bm*(+)—book-to-market; *agr*(−)—asset growth; and *gma*(+)—profitability; as part of the set of active FC. Only *mve* —size; is missing. Many out of the 21 FC selected by the adaptive Lasso are based exclusively on price information. This includes the most relevant FC, measured by the absolute size of the coefficient, *mom1m*(−)—short-term reversal; Moreover, the prominent and classical 12 months momentum—*mom12m*(+) takes the spot of the second-most relevant FC. Moreover, the adaptive Lasso selects the following price-related FC: *idiovol*(−)—last month idiosyncratic return volatility; *maxret*(−)— the max daily return of the previous month; and *idiovol*(−)—last month idiosyncratic volatility.

We skip all other selected FC and refer to Table 3 instead. Furthermore, the table expresses the differences between the three selection specifications and underscores the relevance of this choice. Generally, these across-study comparisons have to be conducted with caution, as the set of FC and the sample periods can differ and hence impact the inference in an unknown way. POLS on the other hand selects a set of 23 FC, whose elements largely overlap with the set identified by the adaptive Lasso.

The table reveals that a Lasso-based procedure would suggest an even higher dimensional relation between FC and returns. We dispense with the discussion here: As the simulation results showed, lasso could be severely affected by false positives and generally overestimates the number of active variables. Overall, we find a substantial number of FC inspected do not contain relevant information for predicting returns when considered in a multivariate selection, as 41 of the included 62 FC are not picked by the adaptive Lasso.

**Table 3. Main pooled FC regression** $r = X'\gamma + \epsilon$**:** The table exhibits the regression coefficients of POLS, Lasso (L) and adaptive Lasso (AL). POLS t-values are displayed in brackets, bold t-values indicate significance after a DFDR adjustment at the 5%-level. In case of POLS, row "# selected" counts FC with *p*-values ≤ 5%, the count for the DFDR equivalent is displayed in brackets. The other columns reflect the count of all non-zero FC coefficients. The estimation spans the period from 1974 to 12–31 until 31 December 2020. The AE: 'All', 'Large', 'Mid' and 'Small' define different sets of stocks: all CRSP/CS stocks, market cap rank 1–1000, 1001–2000 and 2000>, respectively.

| Sample Method | All AL CV5 | All L CV5 | All POLS | Large AL CV5 | Large L CV5 | Large POLS | Mid AL CV5 | Mid L CV5 | Mid POLS | Small AL CV5 | Small L CV5 | Small POLS |
|---|---|---|---|---|---|---|---|---|---|---|---|---|
| # selected | 21 | 47 | 23 (13) | 14 | 38 | 10 (0) | 18 | 50 | 14 (6) | 25 | 54 | 17 (9) |
| absacc | −0.50 | −0.85 | −1.03 (**3.36**) | | | −0.19 (0.75) | | −0.32 | −0.49 (1.82) | −0.55 | −0.97 | −1.21 (**3.07**) |
| acc | | −0.45 | −0.77 (2.32) | | −0.33 | −0.70 (2.40) | | −0.34 | −0.60 (1.81) | | −0.82 | −1.25 (2.82) |
| age | | −0.13 | −0.12 (0.56) | | −0.04 | −0.07 (0.25) | | | 0.03 (0.13) | | 0.32 | 0.43 (1.45) |
| agr | −1.96 | −1.29 | −1.52 (**4.67**) | −0.43 | −0.53 | −0.87 (2.30) | −1.02 | −0.95 | −1.17 (2.37) | −2.22 | −1.79 | −2.05 (**5.37**) |
| beta | 1.12 | 1.39 | 1.58 (2.38) | | 0.02 | 0.31 (0.49) | 1.31 | 1.39 | 1.58 (2.23) | 1.06 | 1.14 | 1.33 (2.03) |
| bm | 1.39 | 1.12 | 1.20 (**3.86**) | | 0.41 | 0.72 (1.94) | | 0.18 | 0.30 (0.71) | 0.29 | 0.48 | 0.52 (1.24) |
| bm_ia | | 0.18 | 0.19 (0.73) | | | −0.20 (0.59) | | 0.07 | 0.11 (0.28) | 0.43 | 0.53 | 0.53 (1.76) |
| cash | 0.48 | 0.82 | 0.82 (1.45) | | 0.25 | 0.35 (0.54) | 0.18 | 0.72 | 0.92 (1.55) | 0.79 | 0.99 | 0.93 (1.55) |
| cashdebt | 0.34 | 0.59 | 0.72 (2.25) | | 0.24 | 0.53 (1.35) | 0.37 | 0.65 | 0.82 (1.90) | 0.54 | 0.79 | 0.95 (2.61) |
| cashpr | | −0.08 | −0.13 (0.69) | | −0.09 | −0.21 (0.99) | | −0.08 | −0.15 (0.76) | | −0.27 | −0.38 (1.50) |
| cfp_ia | 0.42 | 0.52 | 0.50 (2.66) | | | −0.22 (1.00) | | | −0.07 (0.30) | 0.48 | 0.54 | 0.53 (1.68) |
| chato_ia | | 0.43 | 0.52 (**3.11**) | | 0.11 | 0.32 (1.43) | | 0.15 | 0.24 (1.11) | | 0.18 | 0.30 (1.13) |
| chg_mom6m | −1.52 | −1.84 | −2.29 (**2.86**) | −3.00 | −2.75 | −3.57 (2.73) | −2.32 | −1.84 | −2.44 (2.43) | | −0.24 | −0.79 (0.95) |
| chinv | | −0.05 | 0.03 (0.15) | | | 0.16 (0.49) | | −0.16 | −0.09 (0.35) | | −0.10 | −0.02 (0.07) |
| chpm_ia | | | 0.07 (0.48) | | 0.03 | 0.15 (1.00) | | 0.27 | 0.36 (1.91) | | −0.10 | −0.20 (0.87) |
| chshrout | −0.35 | −0.58 | −0.63 (**2.84**) | | | −0.05 (0.21) | | −0.35 | −0.45 (1.72) | −0.43 | −0.61 | −0.65 (1.98) |
| currat | | −0.11 | −1.55 (1.85) | | −0.46 | −1.56 (1.69) | −0.30 | −0.71 | −2.05 (1.94) | | | −1.85 (1.66) |
| depr | | 0.04 | 0.09 (0.40) | | 0.08 | 0.11 (0.44) | | 0.20 | 0.18 (0.71) | | 0.48 | 0.62 (1.94) |
| divi | | | −0.03 (0.21) | | −0.13 | −0.23 (1.90) | | −0.08 | −0.17 (0.99) | | | −0.01 (0.04) |
| divo | | 0.01 | 0.08 (0.57) | | | −0.09 (0.72) | | 0.04 | 0.12 (0.80) | | 0.08 | 0.16 (0.74) |
| dolvol | | | 0.46 (0.60) | | | 0.21 (0.33) | | 0.15 | 0.67 (1.58) | | 0.59 | 1.16 (1.68) |
| dy | −0.64 | −0.82 | −0.87 (2.74) | | −0.29 | −0.43 (1.21) | −0.20 | −0.55 | −0.63 (1.91) | −0.48 | −0.76 | −0.86 (2.40) |
| egr | | −0.18 | −0.21 (1.18) | | −0.31 | −0.30 (1.49) | | | −0.05 (0.22) | | 0.12 | 0.26 (1.00) |
| ep | 1.22 | 1.36 | 1.46 (**4.28**) | 0.47 | 0.69 | 0.74 (2.26) | 0.49 | 0.65 | 0.71 (1.81) | 1.82 | 1.95 | 2.12 (**4.88**) |
| gma | 1.46 | 1.33 | 1.42 (**4.74**) | 0.12 | 0.60 | 0.80 (1.95) | 1.06 | 1.08 | 1.22 (**3.56**) | 1.60 | 1.50 | 1.59 (**4.36**) |
| grcapex | | −0.23 | −0.28 (1.87) | | −0.03 | −0.11 (0.56) | −0.47 | −0.65 | −0.72 (**3.47**) | | −0.31 | −0.36 (1.62) |
| hire | | 0.06 | 0.20 (0.94) | | | 0.08 (0.34) | | 0.24 | 0.40 (1.78) | | −0.19 | −0.25 (0.97) |
| idiovol | −3.93 | −4.08 | −4.34 (**7.03**) | −0.22 | −0.59 | −0.90 (1.84) | −2.11 | −2.13 | −2.41 (**4.51**) | −4.62 | −4.46 | −4.67 (**6.91**) |
| invest | | −0.34 | −0.35 (1.15) | | | 0.10 (0.33) | | −0.02 | 0.05 (0.13) | −0.32 | −0.40 | −0.34 (0.89) |

**Table 3.** *Cont.*

| Sample Method | All AL CV5 | All L CV5 | All POLS | Large AL CV5 | Large L CV5 | Large POLS | Mid AL CV5 | Mid L CV5 | Mid POLS | Small AL CV5 | Small L CV5 | Small POLS |
|---|---|---|---|---|---|---|---|---|---|---|---|---|
| lev | | 0.14 | 0.22 (0.50) | | 0.13 | 0.20 (0.43) | | 0.38 | 0.53 (1.26) | 0.46 | 0.70 | 0.83 (1.40) |
| lgr | | | 0.19 (0.94) | | | 0.33 (1.21) | | | 0.01 (0.04) | | | 0.20 (0.71) |
| maxret | | | 0.19 (0.29) | | | 0.57 (1.03) | | | 0.39 (0.71) | | −0.68 | −0.63 (0.71) |
| mom12m | 2.56 | 2.35 | 2.08 (2.12) | | 0.31 | −0.23 (0.20) | | 0.69 | 0.34 (0.32) | 3.88 | 3.59 | 3.28 (**3.05**) |
| mom1m | −4.23 | −4.19 | −4.27 (**6.63**) | −1.31 | −1.39 | −1.65 (3.03) | −2.18 | −2.14 | −2.26 (**4.04**) | −6.48 | −6.23 | −6.23 (**7.41**) |
| mom36m | | 0.26 | 0.40 (0.94) | | | 0.08 (0.19) | | 0.07 | 0.27 (0.72) | | 0.32 | 0.46 (0.96) |
| mom6m | 1.44 | 1.86 | 2.43 (2.06) | 2.49 | 2.13 | 3.17 (2.02) | 2.96 | 2.20 | 2.95 (2.24) | | 0.35 | 1.04 (0.85) |
| mve | | −0.52 | −1.68 (1.33) | −0.64 | −1.23 | −2.01 (2.43) | −0.18 | −0.53 | −0.97 (2.42) | | −0.09 | −0.70 (0.65) |
| mve_ia | | | 0.54 (0.49) | | 0.48 | 0.95 (1.36) | | 0.19 | 0.37 (0.68) | −0.39 | −0.85 | −0.87 (0.98) |
| pchcapx_ia | | −0.33 | −0.38 (2.38) | | −0.19 | −0.25 (1.53) | | −0.07 | −0.13 (0.87) | −0.17 | −0.44 | −0.49 (2.51) |
| pchcurrat | | | −0.30 (0.66) | | −0.09 | −0.46 (1.00) | | −0.07 | −0.44 (0.86) | | | −0.42 (0.71) |
| pchdepr | | | −0.06 (0.31) | | | −0.02 (0.10) | | | 0.06 (0.27) | | −0.27 | −0.38 (1.46) |
| pchgm_pchsale | | 0.31 | 0.32 (2.05) | | | −0.07 (0.38) | | 0.27 | 0.30 (1.47) | 0.13 | 0.45 | 0.47 (1.80) |
| pchquick | | | 0.42 (0.90) | | | 0.42 (0.88) | | | 0.39 (0.74) | | 0.17 | 0.75 (1.19) |
| pchsale_pchinvt | | 0.24 | 0.29 (1.64) | | | 0.01 (0.05) | | 0.12 | 0.16 (0.77) | | 0.36 | 0.48 (1.83) |
| pchsale_pchrect | | | −0.04 (0.28) | | | 0.02 (0.12) | | −0.08 | −0.20 (1.07) | | 0.35 | 0.46 (2.12) |
| pchsale_pchxsga | | −0.09 | −0.16 (0.70) | | −0.41 | −0.54 (2.33) | −0.18 | −0.51 | −0.62 (2.69) | | | −0.03 (0.09) |
| pchsaleinv | | 0.13 | 0.17 (0.82) | | 0.11 | 0.25 (0.92) | | 0.26 | 0.30 (1.32) | | | −0.16 (0.56) |
| pctacc | | 0.06 | 0.20 (0.73) | | | 0.11 (0.56) | | | −0.01 (0.04) | | 0.18 | 0.37 (1.13) |
| quick | | | 1.46 (1.57) | | | 0.88 (1.05) | | | 1.34 (1.22) | | 0.35 | 2.32 (1.88) |
| rd | | | 0.00 (0.03) | | | −0.11 (1.03) | | 0.21 | 0.25 (1.77) | | −0.09 | −0.19 (0.93) |
| rd_mve | 1.72 | 1.80 | 1.88 (**5.75**) | 0.81 | 0.70 | 0.72 (2.22) | 1.16 | 1.12 | 1.21 (**3.54**) | 2.29 | 2.37 | 2.50 (**6.56**) |
| rd_sale | | 0.09 | 0.23 (0.78) | 0.08 | 0.71 | 1.00 (1.90) | | 0.61 | 0.83 (1.91) | | | 0.09 (0.25) |
| retvol | −2.57 | −2.60 | −2.75 (**3.49**) | −1.24 | −1.08 | −1.62 (2.97) | −2.54 | −2.39 | −2.70 (**4.76**) | −2.54 | −1.94 | −1.96 (1.86) |
| roic | 0.23 | 0.58 | 0.69 (2.68) | 0.72 | 0.83 | 0.84 (1.62) | 0.15 | 0.58 | 0.70 (2.14) | | 0.49 | 0.58 (1.52) |
| salecash | | | 0.07 (0.46) | | | 0.09 (0.53) | | 0.03 | 0.20 (1.12) | | −0.03 | −0.13 (0.51) |
| saleinv | | 0.16 | 0.12 (0.91) | | | −0.13 (0.84) | | | −0.05 (0.28) | | 0.26 | 0.25 (1.02) |
| salerec | | | 0.08 (0.33) | | 0.08 | 0.20 (0.73) | | 0.23 | 0.36 (1.15) | | 0.02 | 0.13 (0.49) |
| sgr | | −0.41 | −0.50 (1.86) | | | −0.21 (0.62) | | | −0.00 (0.00) | −0.50 | −0.66 | −0.72 (2.07) |
| sp | 0.25 | 0.51 | 0.52 (1.58) | 0.23 | 0.48 | 0.52 (1.60) | | 0.24 | 0.24 (0.70) | 0.73 | 0.68 | 0.70 (1.41) |
| tang | | 0.18 | 0.22 (0.68) | −0.02 | −0.46 | −0.60 (1.90) | | −0.09 | −0.26 (0.80) | | 0.09 | 0.12 (0.31) |
| turn | −0.93 | −1.05 | −1.28 (**3.51**) | | −0.32 | −0.59 (1.53) | | −0.50 | −0.97 (2.28) | −2.26 | −2.50 | −2.82 (**4.92**) |

### 5.3.2. FC Selection Conditioned on Size

The selection results conditioned on large-cap stocks only are depicted in the second set of columns in Table 3. Overall, we observe fewer active FC compared to results including all stocks; a total of 14 (vs. 21) are selected. Not overly surprisingly, the top-ranked FC resembles the picture described above, where price-related information dominates the overall prediction contribution—*mom1m*(−), *mom6m*(+) and *chg_mom6m*(−) are among the FC with the highest absolute coefficient values. Moreover, OLS selects a sparser set vs. the adaptive Lasso with ten FC only. The DFDR adjustments suggest none of the included FC are useful in predicting large-cap stocks. Given the out-of-sample prediction results presented above, this reflects the findings that large-cap stocks are not predictable with the linear methods and FC included in this work.

Table 3 also presents the active set of FC conditioned on mid-sized stocks. Strikingly, price-based FC rank highest, with *mom6m*(+), *retvol*(−), *chg_mom6m*(−) and *mom1m*(−) as the top four contributors. Moreover, the full sample estimated mid-cap dimensionality is higher, reflecting once more some predictability for stocks belonging to an economically less relevant segment of the market.

Finally, we briefly discuss the results including exclusively small-cap stocks. The FC selection, as shown in the last block of Table 3, shows that the price information-driven FC is dominant as in the mid-sized selection regression. We find consistency in which FC occupies the top ranks, considering the magnitude of short-term reversal. The three highest rank FC are: *mom1m*(−), *idiovol*(−) and *mom12m*(+). The OLS and Lasso-based selection deviate once more.

## 6. Conclusions

In this work, we propose the application of the adaptive Lasso for predicting cross-sectional stock returns. In particular, this study contributes to a better understanding of the behavior of the adaptive Lasso when applied in panel data settings mimicking the expected returns cross-section dynamics. We perform an extensive Monte Carlo simulation study in which we consider panel data scenarios of low signal-to-noise ratios including heteroscedastic, non-normal and highly cross-sectionally correlated errors. We compare the accuracy of the adaptive Lasso, Lasso and POLS based on the ability to select the truly informative FC for prediction and on the final predictive performance. The selection results show that the Lasso is inferior to its adaptive version in most specifications. In particular, a required condition, most apparent in cases of higher correlations, reveals shortcomings in the Lasso. Despite these apparent selection disadvantages for the standard Lasso, both Lasso-type methods yield improved predictive results over their classical alternatives. The adaptive Lasso appears promising compared to OLS, especially at reducing type II error ratios and controlling FC that suffer from a likely publication bias. POLS-based predictions show the least promising results.

Furthermore, in agreement with the previous literature, we show that the predictability of linear methods based on a rich zoo of firm characteristics is mostly limited to small and micro-cap stocks—the least relevant section of the stock market. Large-cap stocks are not predictable with the linear methods used in this work. Overall, the predictive differences between different linear methods are hard to measure given the potentially non-existing predictability of large-cap stocks. When emphasizing the evaluation based on the less relevant but predictable small and micro-cap segment, we find that Lasso-type predictions perform best. This empirical finding is consistent with the results of the simulation study. An adaptive Lasso selection procedure applied to 62 FC included in this paper and constructed based on US stock data from 1975 to 2020 identifies a highly dimensional return process. We show that a large part of published FC is selected when considered in a multivariate predictive analysis simultaneously; we identify 21 FC of relevance for prediction.

**Author Contributions:** Conceptualization, M.M. and F.A.; methodology, M.M. and F.A.; software, M.M.; validation, M.M.; formal analysis, M.M.; investigation, M.M.; resources, M.M.; data curation, M.M.; writing—original draft preparation, M.M.; writing—review and editing, F.A.; supervision, F.A. All authors have read and agreed to the published version of the manuscript.

**Funding:** This research received no external funding.

**Data Availability Statement:** The data can be obtained from the authors upon request.

**Acknowledgments:** The authors are grateful to workshop participants at the CMStatistics 2016 conference, University of St.Gallen and University of Konstanz. Furthermore, we are thankful for insightful comments from Robert Hodrick, Alberto Martin-Utrera, Juan-Pablo Ortega and Paul Söderlind.

**Conflicts of Interest:** The authors report there are no competing interest to declare.

**Appendix A**

*Appendix A.1. Signal-to-Noise Ratio*

In order to calibrate the simulation to the desired signal-to-noise ratio, we set the volatility of the factors and idiosyncratic volatility as follows. The signal-to-noise ratio (SNR) is defined as:

$$SNR = \frac{\sigma^2_{\text{signal}}}{\sigma^2_{\text{noise}}}$$

and it is related to the r-squared as follows:

$$SNR = \frac{R^2}{1 - R^2} \tag{A1}$$

Recalling Equation (1) and ignoring the indices, we can write,

$$r = x'f + \eta,$$

with $f = \mu + \epsilon$

Hence, we can define the variance of the signal as:

$$\sigma^2_{\text{signal}} = \text{Var}(x'\mu)$$
$$= \sigma^2_x \, \mu'\mu$$

Note that the $\mu$ is defined as a uniformly distributed random vector and the realized $\mu$ are fixed at the beginning of the simulation and can be treated as deterministic.

Furthermore, the variance of the noise can be stated as follows:

$$\sigma^2_{\text{noise}} = \text{Var}(x'\epsilon) + \text{Var}(\eta)$$
$$= \sigma^2_x \, \sigma^2_{\epsilon} + \sigma^2_{\eta}$$
$$= \sum_c^{|\mathcal{C}|} \sigma^2_{\epsilon,c}\sigma^2_x I_{c\in\mathcal{P}\cup\mathcal{U}} + \sigma^2_{\eta}$$
$$= \sigma^2_{\epsilon,1}\sigma^2_x + (P + U - 1)\sigma^2_{\epsilon_f}\sigma^2_x + \sigma^2_{\eta}$$

The first line can be simplified according to Equation (A2) below as all terms involving $\text{Cov}(x,\epsilon)$, $E(\epsilon)$ and $E(x)$ collapse to zero. $\sigma_{\epsilon,1}$ is given by the data reflecting the long-term mean of the stock market GARCH volatility. $\sigma_{\epsilon_f}$ and $\sigma_{\eta}$ are calibrated such that each part contributes equally to fit the desired signal-to-noise ratio ($\sigma^2_{\eta} = (P + R - 1)\sigma^2_{\epsilon_f}$). The value of $\sigma^2_{\eta}$ and $\sigma^2_{\epsilon_f}$ of the desired SNR or the desired $R^2$ follow then straightforwardly.

The variance of the product of two random variables $X$ and $Y$ can be expressed as follows:

$$
\begin{aligned}
\text{Var}(XY) =& E[X^2Y^2] - [E(XY)]^2 \\
=& \text{Cov}(X^2, Y^2) + E(X^2)E(Y^2) - [E(XY)]^2 \\
=& \text{Cov}(X^2, Y^2) + (\text{Var}(X) + [E(X)]^2)(\text{Var}(Y) + [E(Y)]^2) \\
& - [\text{Cov}(X, Y) + E(X)E(Y)]^2
\end{aligned} \tag{A2}
$$

Moreover, we can show that the $R^2$ of a frequency with length $T$ and a frequency comprising a fraction $\frac{T}{\tau}$ of it are related as follows, assuming that $x$ does not change with the time horizon and all terms in $\sigma^2_{\text{noise}}$ are treated as returns with zero auto-correlation:

$$
\sigma^2_{\text{signal},T} = \text{Var}(x'\mu) = \sigma^2_x \, \mu'\mu
$$

$$
\sigma^2_{\text{signal},\frac{T}{\tau}} = \text{Var}(x'\frac{\mu}{\tau}) = \sigma^2_x \, \frac{\mu'\mu}{\tau^2}
$$

$$
\sigma^2_{\text{noise},\frac{T}{\tau}} = \frac{1}{\tau}\sigma^2_{\text{noise},T}
$$

and hence,

$$
\begin{aligned}
\frac{SNR_T}{SNR_{\frac{T}{\tau}}} &= \frac{\sigma^2_{\text{signal},T}}{\sigma^2_{\text{signal},\frac{T}{\tau}}} \frac{\sigma^2_{\text{noise},\frac{T}{\tau}}}{\sigma^2_{\text{noise},y}} \\
&= \frac{\mu^2\sigma^2_x}{\frac{1}{\tau^2}\mu^2\sigma^2_x} \frac{\frac{1}{\tau}\sigma^2_{noise,T}}{\sigma^2_{noise,T}} = \tau
\end{aligned} \tag{A3}
$$

Combining Equations (A3) and (A1), the following relation holds:

$$
\frac{R^2_{\frac{T}{\tau}}}{1 - R^2_{\frac{T}{\tau}}}\tau = \frac{R^2_T}{1 - R^2_T} \tag{A4}
$$

*Appendix A.2. Simulation Study: Additional Robustness Checks*

In addition to the sensitivity analysis provided in Section 3.2 this section exhibits additional robustness checks. The results are presented in Table A1 and Figure A1.

Appendix A.2.1. Additional Cases

**Case A1: Time-constant stock market volatility**
Default settings, except for the assumption of the underlying latent volatility process of $\hat{\sigma}^2_{t,0}$, which we fix for all $t$ to the long-term volatility estimate of the US stocks of 15.8 %.
**Case A2: t-distributed stock market returns**
Default settings, except $\epsilon_{t+1,0} \sim t$-with the GARCH(1,1) estimation also based on the t-distributed errors, with an estimated number of degrees of freedom, $\hat{v}$, of 7.14. However, differences are not overly strong.
**Case A3: Time varying risk-premia**
The model is identical to Model 1 except that it is equipped with a time-varying $\mu_t$, instead of the time constant $\mu$ as before. Notice, that this case can collapse to the first model from a statistical perspective. For example, assume the following time-varying process: $\mu_t = \bar{\mu} + \kappa_t$, and, $\kappa_t \sim \mathcal{N}(0, \sigma_\kappa)$. Any variation in the risk-premia would then be absorbed by the error term and cannot be distinguished from it. In a panel regression, we would then simply estimate $\bar{\mu} = \sum_{t=1}^{T} \mu_t$. Otherwise, default settings. We impose that, $\mathbb{E}[\mu_{t,0}]$

is equivalent to $\mu_0$. The stock market risk-premia is replaced with the following AR(1) process:

$$\mu_{t,c} = c_c + \varphi \mu_{t-1,c} + \psi_t,$$

the constant, $c_c$, is set to $\mu_c(1 - \varphi) = c_c$ for the mean constraint to hold and the noise component, $\psi_t \sim \mathcal{N}(0, \sigma_{\mu_0}^2)$ and $\varphi = 0.2$. The size of the standard error, $\sigma_{\mu_0}$, is set such that it absorbs $1/5$ of the unconditional variance of the associated stock market factor variance (which is reduced accordingly). The errors are independent of each other.

Appendix A.2.2. Results

**Case A1: Time constant stock market volatility**
Table A1 reveals that assuming homoscedastic errors for the stock market factor only marginally affects the error ratios. The only difference we observe is a slight drop in the error ratio for the stock market factor for the OLS approach.

**Case A2: T-distributed stock market returns**
Using draws from a Student-t distribution instead of the Normal with corresponding GARCH-(1,1) volatility estimates for the market factor does not have an impact on the performance behavior of the three different methods, as Table A1 shows.

**Case A3: Time varying risk-premia**
Assuming an AR(1) process for the mean component of the stock market factor does not influence the inference by much. We document only marginal changes for all methods in this case for FC 0.

**Table A1. Simulation Results:** The table provides an overview of type II (for priced factor FC, FC 0–5) and type I error (for unpriced factor FC, 6–11, and spurious FC, 12–99) ratio behavior in percentage points for the specified simulation cases. All details of the cases can be found in Appendix A.2.1. FC cases 12–15 are interesting insofar as they are spurious FC with high correlations to priced and unpriced FC. A white space represents a zero value. Please note that the results displayed are based on 200 simulations only.

| Case | Method | $\mathcal{P}$ — Type II | | | | | | $\mathcal{U}$ — Type I | | | | | | $\mathcal{S}$ — Type I | | | | Summary FC 16–99 | | | |
| | | 0 | 1 | 2 | 3 | 4 | 5 | 6 | 7 | 8 | 9 | 10 | 11 | 12 | 13 | 14 | 15 | mean | std | min | max |
|---|---|---|---|---|---|---|---|---|---|---|---|---|---|---|---|---|---|---|---|---|---|
| 1a | POLS | 0.46 | 0.01 | | | | | 0.05 | 0.06 | 0.04 | 0.03 | 0.05 | 0.06 | 0.05 | 0.03 | 0.06 | 0.07 | 0.05 | 0.02 | 0.02 | 0.10 |
| | POLS BY | 0.84 | 0.10 | 0.01 | | 0.01 | | | | | | | 0.01 | 0.01 | | | 0.01 | | | | 0.01 |
| | POLS Holm | 0.85 | 0.10 | 0.01 | | 0.01 | | | | | | | 0.01 | | | | 0.01 | | | | 0.01 |
| | POLS Bonf | 0.85 | 0.10 | 0.01 | | 0.01 | | | | | | | 0.01 | | | | 0.01 | | | | 0.01 |
| | AL BIC | 0.01 | | | | | | 0.03 | 0.15 | 0.12 | 0.14 | 0.17 | 0.18 | | | | | | | | |
| | AL AIC | 0.01 | | | | | | 0.03 | 0.19 | 0.12 | 0.18 | 0.20 | 0.21 | | | | | | | | |
| | AL CV5 | 0.06 | 0.02 | 0.02 | 0.01 | 0.02 | 0.01 | | 0.08 | 0.06 | 0.05 | 0.07 | 0.09 | | | | | | | | |
| | L BIC | 0.01 | | | | | | 0.41 | 0.66 | 0.62 | 0.70 | 0.66 | 0.76 | 0.24 | 0.15 | 0.06 | 0.02 | 0.01 | 0.01 | | 0.03 |
| | L AIC | | | | | | | 0.56 | 0.80 | 0.84 | 0.91 | 0.84 | 0.91 | 0.34 | 0.32 | 0.28 | 0.32 | 0.25 | 0.03 | 0.18 | 0.33 |
| | L CV5 | 0.03 | 0.02 | 0.01 | 0.01 | 0.01 | 0.01 | 0.41 | 0.54 | 0.52 | 0.60 | 0.57 | 0.59 | 0.22 | 0.14 | 0.04 | 0.02 | 0.02 | 0.01 | | 0.04 |
| 2a | POLS | 0.97 | | | | | | 0.06 | 0.06 | 0.04 | 0.04 | 0.07 | 0.06 | 0.06 | 0.06 | 0.04 | 0.04 | 0.05 | 0.02 | 0.01 | 0.09 |
| | POLS BY | 1.00 | | 0.12 | | | 0.01 | | 0.01 | | 0.01 | 0.01 | 0.01 | 0.01 | 0.01 | | | | | | 0.01 |
| | POLS Holm | 1.00 | | 0.12 | | | 0.01 | | 0.01 | | 0.01 | 0.01 | 0.01 | 0.01 | 0.01 | | | | | | 0.01 |
| | POLS Bonf | 1.00 | | 0.12 | | | 0.01 | | 0.01 | | 0.01 | 0.01 | 0.01 | 0.01 | 0.01 | | | | | | 0.01 |
| | AL BIC | 0.13 | | | | | | 0.02 | 0.14 | 0.12 | 0.20 | 0.17 | 0.15 | | | | | | | | |
| | AL AIC | 0.11 | | | | | | 0.02 | 0.18 | 0.17 | 0.23 | 0.20 | 0.18 | | | | | | | | |
| | AL CV5 | 0.53 | 0.19 | 0.27 | 0.21 | 0.19 | 0.20 | 0.01 | 0.04 | 0.01 | 0.04 | 0.03 | 0.02 | | | | | | | | |
| | L BIC | 0.04 | | | | | | 0.38 | 0.65 | 0.63 | 0.65 | 0.61 | 0.65 | 0.26 | 0.17 | 0.02 | 0.02 | 0.01 | 0.01 | | 0.03 |
| | L AIC | 0.02 | | | | | | 0.48 | 0.79 | 0.84 | 0.83 | 0.84 | 0.88 | 0.38 | 0.34 | 0.34 | 0.23 | 0.25 | 0.03 | 0.18 | 0.31 |
| | L CV5 | 0.30 | 0.10 | 0.14 | 0.11 | 0.10 | 0.11 | 0.24 | 0.35 | 0.33 | 0.31 | 0.31 | 0.33 | 0.14 | 0.09 | 0.03 | 0.02 | 0.02 | 0.01 | | 0.04 |
| 3a | POLS | 0.97 | | | | | | 0.04 | 0.06 | 0.06 | 0.08 | 0.05 | 0.07 | 0.04 | 0.05 | 0.05 | 0.02 | 0.05 | 0.01 | 0.02 | 0.08 |
| | POLS BY | 1.00 | | | | | | | | 0.01 | 0.01 | | | | | | | | | | 0.01 |
| | POLS Holm | 1.00 | | | | | | | | 0.01 | 0.01 | | | | | | | | | | 0.01 |
| | POLS Bonf | 1.00 | | | | | | | | 0.01 | 0.01 | | | | | | | | | | 0.01 |
| | AL BIC | 0.15 | | | | | | | 0.06 | 0.06 | 0.12 | 0.06 | 0.09 | | | | | | | | |
| | AL AIC | 0.15 | | | | | | | 0.07 | 0.08 | 0.14 | 0.10 | 0.11 | | | | | | | | |
| | AL CV5 | 0.44 | 0.11 | 0.11 | 0.11 | 0.11 | 0.11 | | 0.01 | 0.01 | 0.01 | 0.01 | 0.01 | | | | | | | | |
| | L BIC | 0.03 | | | | | | 0.43 | 0.55 | 0.57 | 0.67 | 0.69 | 0.63 | 0.28 | 0.16 | 0.04 | 0.01 | 0.01 | 0.01 | | 0.03 |
| | L AIC | 0.02 | | | | | | 0.56 | 0.74 | 0.80 | 0.89 | 0.86 | 0.83 | 0.41 | 0.34 | 0.21 | 0.23 | 0.23 | 0.03 | 0.14 | 0.33 |
| | L CV5 | 0.28 | 0.08 | 0.09 | 0.08 | 0.09 | 0.08 | 0.28 | 0.29 | 0.33 | 0.33 | 0.40 | 0.34 | 0.12 | 0.12 | 0.03 | 0.01 | 0.01 | 0.01 | | 0.04 |

*Appendix A.3. Firm Characteristics*

The data in the paper utilize the CRSP/Compustat Merged database. In particular, we use the monthly and daily stock data from CRSP Stock/Security files. We calculate weekly returns based on daily data using Friday as the last day of the week. Additionally, factor returns as well as risk-free rate data are obtained from Kenneth French's Data Library. The FC constructed follow the methodology used in Green et al. [3]. Generally, we use yearly accounting data. We construct the single criteria as described in Tables A2 and A3. Some raw data used to construct the FC have data gaps; we follow Green et al. [3] in replacing the missing data points with zeros for the following raw data variables: *xrd, emp, dp, rect, invt, dvt, che, nopi* and *at*.

**Table A2.** The table displays the firm characteristics used. Most definitions are taken from Green et al. [3]. If not otherwise stated, accounting ratios always refer to fiscal year end values.

| ID | Acronym | Name | Description | Reference |
|---|---|---|---|---|
| 1 | beta | Beta | Measured based on 3 years (min 52 weeks) weekly excess returns with standard ols ($y = c + \beta x$) | Fama and MacBeth [8] |
| 2 | beta_sq | Beta squared | Simply obtained by squaring the $\beta$ based on the beta from # 1 | Fama and MacBeth [8] |
| 3 | retvol | Volatility | Volatility is measured by the standard deviation of daily returns of the previous months | Ang et al. [47] |
| 4 | maxret | Maximum return | Maximum return is defined over the max of the daily returns in month $t - 1$ | Bali et al. [48] |
| 5 | idiovol | Idiosyncratic volatility | Calculated based on the residuals of regression in # 1 | Ali et al. [49] |
| 6 | mom1m | 1-month momentum | Return in month $t - 1$ | Jegadeesh [50] |
| 7 | mom6m | 6-month momentum | Cumulative return over 5 months ending in $t - 2$ | Jegadeesh and Titman [51] |
| 8 | mom12m | 12-month momentum | Cumulative return over 11 months ending in $t - 2$ | Jegadeesh [50] |
| 9 | mom36m | 36-month momentum | Cumulative return over 24 months ending in $t - 13$ | Bondt and Thaler [52] |
| 10 | mve | Market capitalization (size) | log of (SHROUT×PRC) | Banz [53] |
| 11 | ep | Earnings-to-price | Earnings per share | Basu [54] |
| 12 | dy | Dividends-to-price | Yearly dividends (dvt) divided by market cap at fiscal year | Litzenberger and Ramaswamy [55] |
| 13 | bm | Book-to-market | Book value of equity (ceq) divided by market cap | Rosenberg et al. [56] |
| 14 | lev | Leverage | Total liabilities (lt) divided by market cap | Bhandari [57] |
| 15 | currat | Current ratio | Current assets (act) divided by current liabilities (lct) | Ou and Penman [58] |
| 16 | pchcurrat | Pct change in current ratio | Percentage change in currat from year $t - 1$ to $t$ | Ou and Penman [58] |
| 17 | quick | Quick ratio | Current assets (act) minus inventory (invt), divided by current liabilities (lct) | Ou and Penman [58] |
| 18 | pchquick | Pct change in quick ratio | Percentage change in quick from year $t - 1$ to $t$ | Ou and Penman [58] |
| 19 | salecash | Sales-to-cash | Annual sales (sale) divided by cash and cash equivalents (che) | Ou and Penman [58] |
| 20 | salerec | Sales-to-receivables | Annual sales (sale) divided by accounts receivable (rect) | Ou and Penman [58] |
| 21 | saleinv | Sales-to-inventory | Annual sales (sale) divided by total invetory (invt) | Ou and Penman [58] |
| 22 | pchsaleinv | Pct change in sales-to-inventory | Percentage change in saleinv from year $t - 1$ to $t$ | Ou and Penman [58] |
| 23 | cashdebt | Cashflow-to-debt | Earnings before depreciation and extraordinary items (ib + dp) divided by avg total liabilities (lt) | Ou and Penman [58] |
| 24 | baspread | Illiquidity (bid-ask-spread) | Monthly avg of daily bid-ask spread divided by avg of daily bid-ask spread | Amihud and Mendelson [59] |
| 25 | depr | Depreciation-to-gross PP&E | Depreciation expense (dp) divided by gross PPE (ppegt) | Holthausen and Larcker [60] |
| 26 | pchdepr | Pct change in Depreciation-to-gross PP&E | Percentage change in depr from year $t - 1$ to $t$ | Holthausen and Larcker [60] |
| 27 | mve_ia | Industry-adjusted firm size | Log market caps are adjusted by log of the mean of the industry | Asness et al. [61] |
| 28 | cfp_ia | Industry-adjusted cashflow-to-price | Industry adjusted cash flow-to-price ratio equal weighted average | Asness et al. [61] |
| 29 | bm_ia | Industry-adjusted book-to-market | Industry adjusted book-to-market equal weighted average | Asness et al. [61] |
| 30 | sgr | Annual sales growth | Percentage change in sales from year $t - 1$ to $t$ | Lakonishok et al. [62] |
| 31 | ipo | IPO | Indicated by 1 if first 12 months available on CRSP monthly file | Loughran and Ritter [63] |
| 32 | divi | Dividend initiation | Indicated by 1 if company pays dividends but did not in prior year. | Michaely et al. [64] |
| 33 | divo | Dividend omission | Indicated by 1 if company does not pay dividends but did in prior year. | Michaely et al. [64] |
| 34 | sp | Sales-to-price | Annual sales (sale) divided by market cap | Barbee Jr et al. [65] |
| 35 | acc | WC accruals | (ib) - (oancf)/(at), if (oancf) is missing then (ib)-(delta_act)-(delta_che) -(delta_lct) + (delta_dlc) + (txp-dp) where each item 0 if missing | Sloan [66] |
| 36 | turn | Share turnover | Average monthly trading volume for the three months $t - 3$ to $t - 1$ divided by SHROUT at $t - 1$ | Datar et al. [67] |
| 37 | pchsale_pchinvt | Delta pct change sales vs. inventory | Difference of percentage changes in sales (sale) and inventory (invt) | Abarbanell and Bushee [68] |
| 38 | pchsale_pchrect | Delta pct change sales vs. receivables | Difference of percentage changes in sales (sale) and receivables (rect) | Abarbanell and Bushee [68] |
| 39 | pchcapx_ia | CAPEX | Industry adjusted (two digit SIC) fiscal year mean adjusted percentage change in capital expenditures (capx) | Abarbanell and Bushee [68] |
| 40 | pchgm_pchsale | Delta pct gross margin vs. sales | Annual percentage change in gross margin (sale minus cogs) minus percentage change in sales (sale) | Abarbanell and Bushee [68] |

**Table A3.** The table displays the firm characteristics used. Most definitions are taken from Green et al. [3]. If not otherwise stated, accounting ratios always refer to fiscal year end values.

| ID | Acronym | Name | Description | Reference |
|---|---|---|---|---|
| 41 | pchsale_pchxsga | Delta pct sales vs. SGaA | Annual percentage change in sales (sale) minus percentage change in SGaA (xsga) | Abarbanell and Bushee [68] |
| 42 | dolvol | Dollar trading volume | Log of trading volume times price per share from month t-2 | Chordia et al. [69] |
| 43 | std_dolvol | Volatility trading volume | Monthly standard deviation of daily trading volume | Chordia et al. [69] |
| 44 | std_turn | Volatility turnover | Monthly standard deviation of daily share turnover | Chordia et al. [69] |
| 45 | chinv | Change in inventory | First difference of inventory (invt) divided by total assets | Thomas and Zhang [70] |
| 46 | pchemp_ia | Industry-adjusted pch in employees | Industry adjusted percentage change in employees | Asness et al. [61] |
| 47 | cfp | Cashflow-to-price | Operating cash flows (oancf) scaled by market capitalization (fiscal year end) | Desai et al. [71] |
| 48 | rd | R&D Increase | If annual increase in R&D expenses (xrd) scaled by total assets (at) >0.05, 1, else 0 | Eberhart et al. [72] |
| 49 | lgr | Pct change in long-term debt | Annual percentage change in long term debt (lt) | Richardson et al. [73] |
| 50 | egr | Pct change in book equity | Annual percentage change in book equity (ceq) | Richardson et al. [73] |
| 51 | rd_sale | R&D-to-sales | R&D expenses(xrd) scaled by sales (sale) | Guo et al. [74] |
| 52 | rd_mve | R&D-to-market | R&D expenses(xrd) scaled by market cap | Guo et al. [74] |
| 53 | chg_mom6m | change in mom6m | difference of mom6m measured at $t$ and $t-6$ | Gettleman and Marks [75] |
| 54 | hire | Pct change in employee | Annual percentage change in employee (emp) | Belo et al. [76] |
| 55 | agr | Asset growth | Annual percentage change in assets (at) | Cooper et al. [77] |
| 56 | cashpr | Cash productivity | Market cap plus long term debt (dltt) minus assets (at) divided by cash (che) | Chandrashekar and Rao [78] |
| 57 | gma | Gross-profitability | Sales (sale) minus costs of goods sold (cogs) divided by one-year lagged assets(at) | Novy-Marx [79] |
| 58 | cash | Cash-to-assets | Cash (che) divided by assets(at) | Palazzo [80] |
| 59 | pctacc | Accruals-to-income | (ib) minus (oancf) divided by abs ((ib)), when (ib) equals 0, it is set to 0.01, if (oancf) is missing then (ib)-(delta_act)-(delta_che) -(delta_lct) + (delta_dlc) + (txp-dp) where each item 0 if missing | Hafzalla et al. [81] |
| 60 | absacc | Absolut accruals | Absolute value of acc | Bandyopadhyay et al. [82] |
| 61 | roic | Return on invested capital | Earnings before interest and taxes (ebit) - non-operating income (nopi), divided by non-cash enterprise value (ceq+lt-che) | Brown and Rowe [83] |
| 62 | grcapex | Pct change in two year CAPX | Percentage change in two year capital expenditure (capx) | Anderson and Garcia [84] |
| 63 | tang | Debt capacity-to-firm-tangability | (Cash (che) + 0.715 receivables (rect) + 0.547 inventory(invt) + 0.535 (ppegt))/ total assets (at) | Hahn and Lee [85] |
| 64 | chshrout | Change in shares-outstanding | Yearly percentage change in outstanding shares (SHROUT) | Pontiff and Woodgate [86] |
| 65 | invest | CAPEX and inventory | Yearly difference in gross property, plant and equipment (ppegt) + diff in (invt) / (t-1) total assets (at) | Chen and Zhang [87] |
| 66 | age | Years since CS coverage | Years since first compustat coverage years(datadate - min(datadate)) | Jiang et al. [88] |
| 67 | chpm_ia | Industry-adjusted change in profit margin | Industry adjusted (two-digit SIC) change in profit margin (ib/sale) | Soliman [89] |
| 68 | chato_ia | Industriy-adjusted change in asset turnover | Industry adjusted (two-digit SIC) change in asset turnover (sale/at) | Soliman [89] |

*Appendix A.4. Code*

Our code is available upon request via github.com; please make requests by email. It is all written in Python 3.x and should be compatible on win and ux systems. Be aware the simulations as specified above are memory/RAM intensive; in order to run the main simulation, at least 90GB of available RAM are required.

*Appendix A.5. Additional Figures*

**Figure A1.** The figure shows the simulated out-of-sample prediction evaluation for each method. Pooled OLS (POLS), Lasso AIC with post-variable selection OLS (L AIC PVOLS), pooled OLS with post-variable selection OLS (POLS PVOLS), Lasso CV5 with post-variable selection OLS (L CV5 PVOLS), adaptive Lasso CV5 (AL CV5), Lasso BIC with post-variable selection OLS (L BIC PVOLS), Lasso AIC (L AIC), adaptive Lasso BIC (AL BIC), adaptive Lasso CV5 with post-variable selection OLS (AL CV5 PVOLS), adaptive Lasso AIC (AL AIC), adaptive Lasso AIC with post-variable selection OLS (AL AIC PVOLS), pooled OLS with post-variable selection OLS and the BHY t-value adjustment(POLS PVOLS BHY), pooled OLS with post-variable selection OLS and the Bonferroni t-value adjustment (POLS PVOLS Bonf), pooled OLS with post-variable selection OLS and the Holm t-value adjustment(POLS PVOLS Holm), adaptive Lasso BIC with post-variable selection OLS (AL BIC PVOLS), Lasso CV5 (L CV5) and Lasso BIC (L BIC). The plot on the left side shows average MCS p-values over 200 simulation cases, where the p-value measures if the model is part of the MCS. The right figure illustrates the MSE values relative to the max of each case. Please note that the results displayed are based on 200 simulations only.

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
