# Peer review of "The Lasso and the Factor Zoo-Predicting Expected Returns in the Cross-Section"

_forecasting, doi:10.3390/forecast4040053_

Round 1
Reviewer 1 Report
- The paper poses a very interesting question and manages to answer it in an outstanding manner.
I suggest the publication of this paper as is for the following reasons:
- highly suited for the journal specifics
- strong rooted into current relevat literature on all aspects of the analysis (factor modeling, Fama-French approach, Fama-McBeth modeling framework, robust OLS coefficient estimates, prediction quality metrics, setup of the simulation study)
- performance evaluation - the different impact on size is interesting. It is not the intent of the paper to produce such answers, but it would be interesting to comment on the economic explanations for this result
- the empirical analysis is relevant and statistical tests are developed in closed connection with the paper's objectives
- Some questions:
- have you tried any non-linear approaches? Given that you used Python libraries, how about SHAP approach to investigate FC as features?
- adding nonlinearity by using squared values of some factors would yield different results?
- what kind of non-linear modeling would be necessary to capture the size effect. I.e. what specification could also allow for the prediction to work in case we would also like to capture the big cap companies?
- All relevant elements of an empirical research analysis are covered.
Author Response
Thank you for the comments. It is important to say that we constrain our selection and prediction procedure to the linear setting because we want to perform a multivariate FC selection based on the character of the FC as proposed in the original publications. So, we did not try non-linear approaches in this study.
However, one of the authors contributed to the fast-growing strand of the literature addressing the prediction problem from a non-linear perspective: Messmer (2017) studies in detail how deep learning can be applied to predict cross-sectional returns. The study by Moritz and Zimmermann (2016) introduces the application of tree-based sorts to achieve better prediction outcomes. Gu et al. (2020) provide a comparison of various non-linear machine learning techniques. For results in the non-linear setting we refer to the above mentioned studies in the introduction, page 4, end of second paragraph.
Reviewer 2 Report
Summary: The authors analyze whether shrinkage estimators (Lasso, adaptive-Lasso) can improve the predictive accuracy when selecting this way the firm characteristics relevant for forecasting the future cross-section of returns. They first provide the results of a simulation study and second apply the competing methods to US stock market data.
Comments: In recent years, many scientific proposals tried to tame the factor zoo with various tools from Machine Learning and/or classical statistical modelling. Thus, I generally find the topic of this paper interesting and as such, I am very sympathetic toward the idea of the paper. However, I have some issues with some elements of the paper that need to be addressed. My specific comments are the following.
- The authors mention the double Lasso of Feng et al. (2020) in the Introduction. Wouldn't it have been obvious to include this method in the simulation study and to investigate its out-of-sample predictive power as well?
- I’m not convinced whether the discussion of type-I or type-II errors is helpful at all for prediction. I just use the best model (eg. in terms of oos-predictive power). Why should I bother with the fact that I have included an FC which is not relevant/ excluded an FC which is not relevant (maybe it proxies other important information.)
- Is there an intercept in eq. (4) to (6), i.e. what is the performance compared to the historical average? Can OLS, Lasso, and adaptive Lasso beat the historical average return?
- It is hard to relate the specified econometric models for the returns to the actual estimated ones. Plugging in (2) in (1), we have an identification problem. Then considering eq. (3), are you really estimating an unconditional mean of the returns? What are the assumptions on the error terms? Did you check them somehow? Please specify the model which is finally estimated in its full beauty.
- What links the simulation study to the final application of real data? What do we learn from the simulation study that is helpful for the empirical application? Do we have a preferred method now? Do we need the application at all?
- Leeb and Pötscher (2005) show that model selection has an important impact on subsequence inference and that ignoring the model selection step leads to invalid inference. Now OLS and Lasso-type estimators are what they call post-model-selection estimators. You use them as well. Does it have any impact on your results?
Author Response
Thank you for taking the time to carefully read our paper and for all your comments. Below the answers to your questions/comments (in italic).
- The authors mention the double Lasso of Feng et al. (2020) in the Introduction. Wouldn't it have been obvious to include this method in the simulation study and to investigate its out-of-sample predictive power as well?
Reply:
Double Lasso methods have been introduced with the goal of making valid in-sample inference for lasso estimators and identifying true casual relations among the covariates and the dependent variable. As we discuss in the Introduction on pages 2-3 the goal of our study is different: The whole emphasis is put on out-of-sample prediction. With this in mind, we think that considering a variant of the lasso introduced to make correct classical in-sample inference does not fit the final goal of our study. Why should such a lasso estimator bring some advantage in term of forecasting accuracy? This is the reason why we did not consider it and still believe it should not be added to the (already large) set of alternative competitors.
Note also that the discussion about the active variables both in the simulations and in the application focuses completely on significance for (out-of-sample) prediction and not in term of stating a true (in-sample) casual relation.
- I’m not convinced whether the discussion of type-I or type-II errors is helpful at all for prediction. I just use the best model (eg. in terms of oos-predictive power). Why should I bother with the fact that I have included an FC which is not relevant/ excluded an FC which is not relevant (maybe it proxies other important information.)
Reply:
On the one side we do not know a-priori what impact could have the inclusion of an insignificant variable and/or the exclusion of an active one on the resulting model’s forecasting accuracy. This is the reason why we perform the whole simulation exercise in an attempt to clarify this point. In fact, we are well aware of the deterioration of the predictive performance that overfitting (estimating the noise for example by including insignificant variables) can lead. We believe that the results of the simulations illustrate well differences in the out-of-sample performance among the considered methods and try to give an interpretation of the loss in performance in term of type I and type II error characteristics of the estimators.
On the other side results of the simulations shed some lights on the reliability of the firm characteristics that are found to be relevant/active for prediction, a question that is highly debated in the current financial literature. Say that the estimator we are using and providing accurate out-of-sample predictions (measured for example using the MSE) choose momentum as active variable. Knowing at least approximately the proportions of type I errors (false positives) the estimator suffers tells us something about the reliability of this result. We now highlight this more clearly at the beginning of Section 5.3 on page 25 and in Section 5.3.1 (related to the Lasso results).
- Is there an intercept in eq. (4) to (6), i.e. what is the performance compared to the historical average? Can OLS, Lasso, and adaptive Lasso beat the historical average return?
Reply:
As common in the asset pricing literature we consider as simple benchmark for the application the naïve historical return average that in our case is the zero-return forecast given that all series are standardized (footnote 3 on page 8 and first paragraph on page 20). Due to the standardization of the dependent variable we do not need any intercept in the different models. As we report in Table 2 the zero-return forecast is consistently excluded from the MCS (see discussion on page 23) of all stocks and small and micro stocks, and therefore outperformed by at least one of the competitors. This result corroborates previous evidence presented in the literature.
- It is hard to relate the specified econometric models for the returns to the actual estimated ones. Plugging in (2) in (1), we have an identification problem. Then considering eq. (3), are you really estimating an unconditional mean of the returns? What are the assumptions on the error terms? Did you check them somehow? Please specify the model which is finally estimated in its full beauty.
Reply:
In (3) we are estimating a conditional mean: Thank you for noticing the mistake. As suggested in the revised version we clarify the model derivation by rewriting it in a more explicit way and highlighted the model assumptions, in particular under the different asset pricing model interpretations; see the revised Section 2.2 on pages 6-8. The assumptions stated for the error terms are the standard ones introduced in factor asset pricing models and have been tested in the previous literature: (i) each idiosyncratic noise component is assumed to be orthogonal to the factor and other idiosyncratic noises, normally distributed with mean zero and constant variance; (ii) the factor noises are assumed to be independent, normally distributed with mean zero and constant variance. Note that these assumptions are relaxed for the market factor in favor of a more realistic fat-tailed factor noise distribution with time-varying variance.
- What links the simulation study to the final application of real data? What do we learn from the simulation study that is helpful for the empirical application? Do we have a preferred method now? Do we need the application at all?
Reply:
This comment is related to the first one above. In the answer we highlighted the existing link between simulation results and the real data application and the reason why simulations are useful to understand and interpret the results of the application that are clearly what the financial community is interested in. The simulation DGP is chosen to be as close as possible to the reality (choice of the parameters) and is specified in accordance with the standard models introduced in asset pricing. As we highlight for example on the top of page 24 the empirical results are mostly consistent with the simulation ones and identify the same best performing models (lasso/ adaptive lasso with CV if, for example, false positives are a concern).
- Leeb and Pötscher (2005) show that model selection has an important impact on subsequence inference and that ignoring the model selection step leads to invalid inference. Now OLS and Lasso-type estimators are what they call post-model-selection estimators. You use them as well. Does it have any impact on your results?
Reply:
Performing valid inference is not a concern in our study given that our focus is entirely on prediction and not on making post-lasso inference. In fact, we never report standard errors for the parameters estimated under the different lasso approaches and simply distinguish between variables been selected or not and their sign (not the size of the effects).
Reviewer 3 Report
see the report

Author Response
Thank you for the suggestion. The mentioned tests for equal predictive ability are tests for pairwise comparisons and do not take into account the inherent multiple testing structure if the predictions of more than two models are considered. As discussed in Section 2.5 on page 12 we test the statistical significance of differences in models’ predictions using the model confidence set (MCS) idea introduced by Hansen that is nowadays the standard approach used to test the hypothesis of superior predictive ability in the case of multiple comparisons. In fact, the MCS is a generalization of the Diebold and Mariano test in a multiple setting and therefore the right approach to use in our case. Results of MCS are reported in Figures 2 and 5 (simulations) and Table 2 (empirical application).
Reviewer 4 Report
Thanks for the fine paper. We all desire sparsity, most notably for its interpretability merits. However, the true DGP may have other plans, and prefer a dense combination of factors. I think the paper would make a much more convincing point by including a ridge regression in the benchmarks (one tuned by GCV, the other by CV5). I fully understand that sparsifying the zoo of factors is the main goal here, but we may only indulge in that only if R^2 gains are not left on the table. The OLS benchmark is certainly dense, but not regularized, and hence severely handicapped. Even if Lasso does ~equally well to Ridge, the authors can still argue that Lasso provides interpretability and is easier to implement for practitioners (fewer factors to gather data on). Nonetheless, we want to know where Ridge and Lasso stand.
Author Response
Thank you for the suggestion. We clearly share your comment about the fact that we do not know for real data applications whether the true data generating process is sparse or dense and, in the second case, whether assuming sparsity can significantly worsen the results. To clarify this point we included the discussion in Section 2.4: As we report there, previous results showed that if the goal is prediction, wrongly assuming sparsity when a large number of covariates is considered does not harm asymptotically the results for lasso type estimators. Moreover, under these conditions results of Ridge and Lasso regressions are very similar, sometimes Lasso predictions even outperform Ridge ones. This is the reason why we did not report results using Ridge regressions: Consistently with the previous empirical evidence we did not notice any particular improvement using Ridge in term of out-of-sample accuracy. Now we report this result in footnote 5 on page 11.
Round 2
Reviewer 3 Report
no more comments